# Associative learning via eyeblink conditioning differs by age from infancy to adulthood
Carolin Konrad [1,2] ✉, Lina Neuhoff[1,2], Dirk Adolph[1,2], Stephan Goerigk[3,4], Jane S. Herbert [5], Julie Jagusch-Poirier[6], Sarah Weigelt [6], Sabine Seehagen [2,7] & Silvia Schneider [1,2] ✉

Associative learning is a key feature of adaptive behaviour and mental health, enabling individuals to adjust their actions in anticipation of future events. Comprehensive documentation of this essential component of human cognitive development throughout different developmental periods is needed. Here, we investigated age-related changes in associative learning in key developmental stages, including infancy, childhood, adolescence, and adulthood. We employed a classical delay eyeblink conditioning paradigm that consisted of two sessions with a total of 48 paired trials. Our initial hypothesis was that performance in associative learning would increase linearly with age. However, our findings suggest that performance peaks during the primary school years: Children in this age-group exhibited superior performance compared to all other age-groups and displayed the most consistent and least variable learning. Adults and adolescents exhibited faster association learning than infants. An additional learning session supported learning in infants and adolescents indicating that during these developmental stages, consolidation processes are vital for learning. A comprehensive account of the development of associative learning may inform theories on aetiology and treatment options in clinical psychology and neurosciences.

Associative learning is a fundamental process by which individuals form connections between stimuli. In its narrowest definition, associative learning describes the encoding of a stimulus or response in relation to another stimulus. Two principal forms of associative learning exist: classical conditioning and operant conditioning[1]. Associative processes are crucial for learning and memory from an early age and play a crucial role in human development and mental health[2,3]. During infancy, infants learn to associate cues, such as parental presence or a familiar object, with feelings of safety. Among other things, early associative learning forms the foundation for emotional regulation[4]. As children mature, they continue to rely on associative learning to acquire new knowledge, skills and social behaviours. For instance, learning to associate specific actions with positive outcomes, such as praise or rewards, reinforces desired behaviours. Thus, associative learning can serve as a key mechanism to achieve numerous complex forms of cognitive operations, behaviour and for mental health[2,5–8].

A classical conditioning task that is a non-invasive tool to study associative memory development from birth onwards is eyeblink conditioning[9–11]. During delay eyeblink conditioning, a tone is paired with an air puff (unconditioned stimulus) delivered to the eye, which triggers an eyeblink (unconditioned response). Repeatedly pairing the tone and the air puff produces a conditioned response: the closure of the eyelid when hearing the tone. Eyeblink conditioning offers some unique advantages. It does not require a complex behavioural or verbal response, making it suitable for studying all age-groups and in particular infants. Furthermore, eyeblink conditioning has a highly conserved neural circuitry and is dependent on the cerebellum[12], offering a window into early brain development. Beyond studying associative learning, eyeblink conditioning research holds potential clinical value. Performance in delay eyeblink conditioning tasks is considered to indicate activity of neural structures implicated in a diverse range of psychopathologies, including attention deficit hyperactivity

[1]Clinical Child and Adolescent Psychology, Mental Health Research and Treatment Center, Faculty of Psychology, Ruhr University Bochum, Bochum, Germany. [2]German Center for Mental Health (DZPG), Partner Site Bochum/Marburg, Marburg, Germany. [3]Department of Psychiatry and Psychotherapy, LMU University Hospital, LMU Munich, Munich, Germany. [4]Charlotte Fresenius Hochschule, Munich, Germany. [5]Wollongong Infant Learning Lab, School of Psychology and Early Start, University of Wollongong, Wollongong, NSW, Australia. [6]Research Unit Vision, Visual Impairments & Blindness, Department of Rehabilitation Sciences, TU Dortmund University, Dortmund, Germany. [7]Developmental Psychology, Faculty of Psychology, Ruhr University Bochum, Bochum, Germany. ✉e-mail: carolin.konrad@rub.de; silvia.schneider@rub.de

disorder, foetal alcohol syndrome, autism spectrum disorder and anxiety disorders[7,13–21].

Associative learning is a form of implicit learning, meaning learning takes place without intention or awareness. The developmental trajectory of implicit learning remains debated—with some studies supporting an 'invariant hypothesis' of no change t[22–24] and others suggesting age-related improvements[25,26]. Regarding operant conditioning, there are linear age-related increases in the rate of learning during the first years of life[27–29]. Research is needed on human developmental work focusing on adolescence in the context of normative operant conditioning[30]. In rodents, there are mixed results when comparing the performance of adolescents and adults[31,32].

Regarding delay eyeblink conditioning, behavioural data suggest that performance changes with age[33–36]. Newborns, 4- and 5-month-olds showed successful acquisition in delay eyeblink conditioning[9,34,35]. Five-month-old infants learned more slowly than adults, but reached the same learning asymptote after a sufficient number of trials[36]. During middle childhood and adolescence, the rate of learning generally increases[37,38]. However, there are mixed results regarding the performance of children compared to adults. Some studies showed that adults produced more conditioned responses compared to 4–13-year-old children[37–39], whereas others found that the conditioned responses were learned at a similar rate in children (9–11), adolescents (17–19 years) and adults[40]. Interpretation of studies on age-related changes in eyeblink conditioning is impaired by differences in methodology across studies and considerations of limited age-ranges within single studies (e.g., no available data for the ages 1–3 years). Consequently, studies show mixed results regarding the performance of children compared to adults. Some reveal advantages for adults[37–39], whereas others show no evidence for differences in performance[40]. Given its fundamental importance for human behaviour and mental health, the absence of a comprehensive record of associative learning across development is thus striking.

Overcoming methodological heterogeneity is pivotal to successfully establishing a comprehensive record of associative learning over the lifespan. Applying the same paradigm to infants as to older populations, however, presents challenges due to ethical constraints and developmental changes in interests, attention rates, and verbal and physical abilities[41]. To address these issues, we previously developed a delay eyeblink conditioning paradigm, which can be utilised from infancy to adulthood[11]. In the present study, we aimed to determine learning curves in a large sample of infants (12-months, $n = 24$; 18-months, $n = 26$; 24-months, $n = 30$; 36-months, $n = 28$), primary school-aged children (age range 7–8 years, $N = 28$), adolescents (age range 12–17 years, $N = 30$) and adults (age range 18–29 years, $N = 64$). We predicted that all age-groups would show successful learning. Based on behavioural data so far, we hypothesised that learning speed (as evidenced by steeper learning curves) would improve from infancy to adulthood[34–36,38]. We expected age differences in acquisition rates such that adults would acquire the association between the tone and the air puff faster than infants[36]. We expected faster acquisition in infants with older age from 12 to 36 months[38]. Lastly, we expected age differences in acquisition rates with adults acquiring the association between the tone and the air puff faster than adolescents. Furthermore, we examined gender differences in acquisition since some studies revealed that girls/women produce a higher number of conditioned responses during classical eyeblink conditioning than boys/men[38]. In light of the considerable heterogeneity of findings and the substantial interindividual differences observed in the extant literature, our additional objective was to identify discrete learning trajectories across age-groups.

## Methods
### Participants
In total, 200 infants, 30 primary school children, 31 adolescents and 65 adults participated in the study. Of all infants, $n = 92$ had to be excluded from the study due to fussiness ($n = 34$), refusal to wear the headband ($n = 52$), falling asleep ($n = 1$), sickness resulting in the cancellation of the second appointment ($n = 3$), issues coordinating the second appointment

($n = 1$) and the COVID-19 pandemic resulting in lockdown and the cancellation of the second appointment ($n = 1$). Therefore, the final infant sample consisted of 108 participants from four different age-groups. The sample included 24 twelve-month-olds ($M_{age} = 362$ days, $SD = 12$, $n = 10$ girls, $n = 14$ boys), 26 eighteen-month-olds ($M_{age} = 547$ days, $SD = 12$, $n = 12$ girls, $n = 14$ boys), 30 twenty-four-month-olds ($M_{age} = 731$ days, $SD = 14$, $n = 16$ girls, $n = 14$ boys) and 28 thirty-six-month-olds ($M_{age} = 1094$ days, $SD = 14$, $n = 14$ girls, $n = 14$ boys). Two primary school children had to be excluded from the study due to sickness resulting in the cancellation of the second ($n = 1$) or the third appointment ($n = 1$). Therefore, the final sample consisted of 28 primary school children ($M_{age} = 7.6$ years, $SD = 0.50$, $n = 14$ girls, $n = 14$ boys). One adolescent had to be excluded from analyses due to a technical failure. Thus, the final sample of adolescents consisted of 30 participants, aged between 12 and 17 years ($M_{age} = 15$ years, $SD = 2$, $n = 16$ girls, $n = 14$ boys). One adult had to be excluded because of scheduling coordination issues. Thus, the final sample of adults consisted of 64 participants, aged between 18 and 29 years ($M_{age} = 21$ years, $SD = 2$, $n = 44$ women, $n = 20$ men). The number of men/boys and women/girls did not differ significantly between age-groups ($\chi^2_{(5)} = 9.72$, $p = 0.084$). Gender was determined using self-report in adolescents and adults, and parent-report in infants and primary school children.

Data was collected until at least 24 participants per age-group were attained or until May 2021 according to the stopping rule. Data collection occurred from December 2017 to May 2021. The achieved power of the interaction effect in the main analysis calculated using G*Power[42] was 1.00. The study was preregistered prior to data analysis on osf: https://osf.io/cvnge. The other preregistered hypotheses on extinction and renewal will be covered in a different manuscript.

Part of the sample has been reported as one group ($N = 57$) in a feasibility study that did not report data on individual age-groups in Konrad et al.[11]. The study was approved by the Ethics Committee of the Department of Psychology at Ruhr University and conducted in accordance with the Declaration of Helsinki and the German Federal Data Protection Act. All adult participants and caregivers of underage participants provided written informed consent prior to the study. Families of infants and school-aged children were recruited using local birth registers. Infants participated within two months of their respective birthdays (12-, 18-, 24-, or 36-months). Primary school children participated from 7 years 0 months to 8 years 11 months. Exclusion criteria in children were hearing or vision problems or preterm birth. Each family was given a payment of 10 € and a small present for the child per session. Recruitment for adult participants was carried out among university students via listserv and word of mouth. Eligibility criteria for participation were being aged between 18 and 30 years. We recruited teenage participants from local events and through informal networks. They were eligible to participate if they were aged between 12 and 17 years old. Participants with slight visual impairments were able to wear glasses during the study and were not excluded. However, wearing contact lenses was not allowed, as it could diminish the participants' sensitivity to the air puff stimulus. The exclusion criteria included hearing problems, neurological disorders, or colour blindness. Adult students received course credit, or alternatively, gift cards worth 10 € per session. Adolescents received gift cards worth 10 € per session.

### Eyeblink paradigm
A comprehensive explanation of the existing 3-day-eyeblink paradigm can be found elsewhere[11]. The paradigm consisted of three session within 14 days. Within the scope of this article, we will only report data on the acquisition session (session one and two).

A mildly aversive air puff to the eye (1/20 lb/in2) served as unconditioned stimulus (US), and a 1 kHz 80 dB tone as conditioned stimulus (CS). The delivery of the air puff was through a headband equipped with a flexible tube. The CS was presented to infants and primary school children through two 8-ohm speakers located at their ear level. CS was administered through headphones in both adolescents and adults.

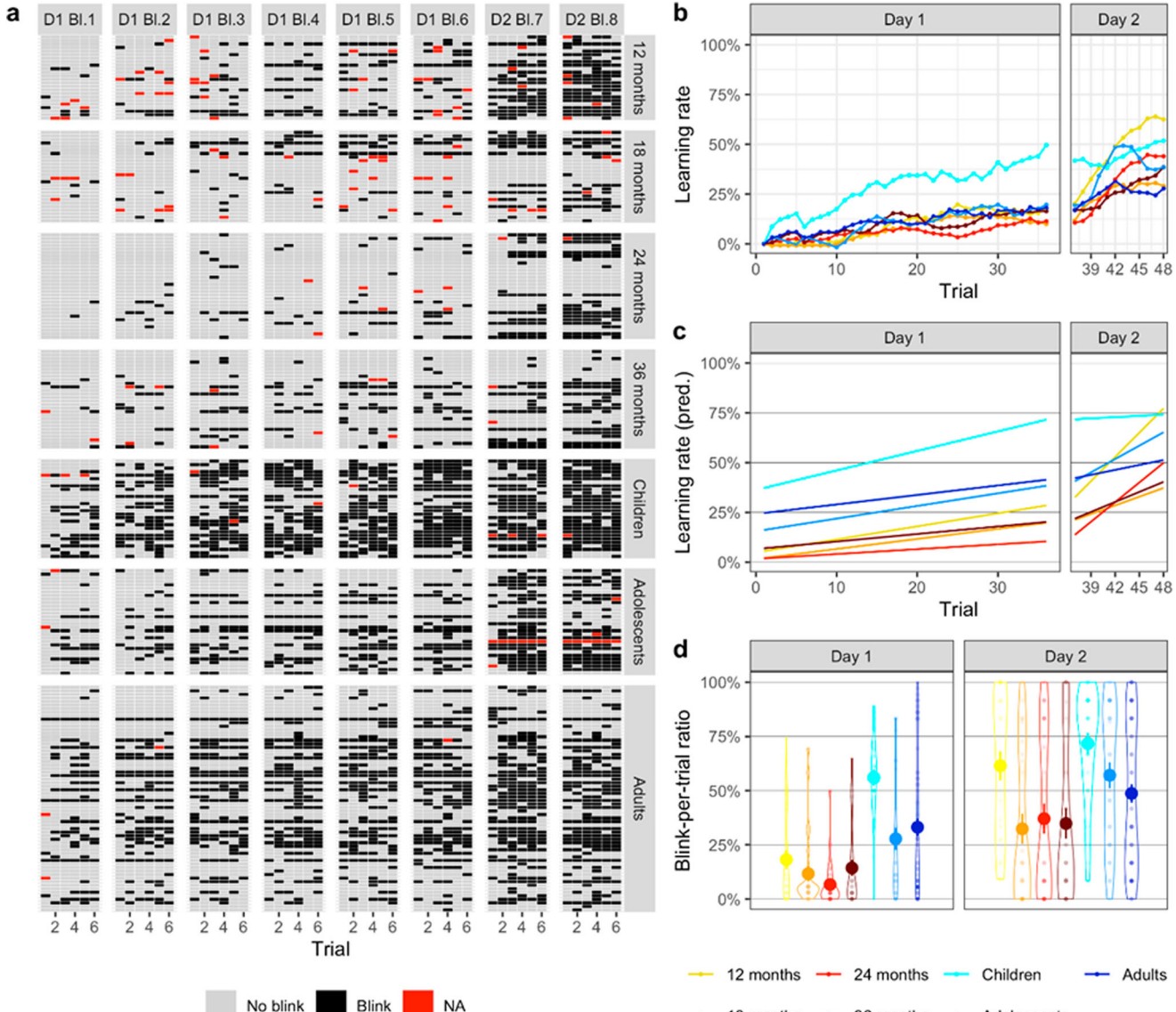

**Fig. 1 | Conditioned responses on day one and two per age-group. a** Conditioned responses for each trial (black squares: blink yes/grey squares: no/ red squares: missing) and each participant in each age-group. **b** Mean conditioned responses per trial on day 1 and 2 for each age-group as defined by ratio of blinks vs. no blinks in a window of 5 trials. Visualisation is baseline centred (yellow line: 12 month-olds ($n = 24$), bright red line: 18-month-olds ($n = 26$), orange line: 24-month-olds ($n = 30$), brown line: 36-month-olds ($n = 28$), turquoise line: primary school children ($n = 28$), light blue line: adolescents ($n = 30$), dark blue line: adults ($n = 64$)). **c** Linear daily increase in conditioned responses as predicted by piecewise linear mixed model parametrised for 2 separate phases (day 1 and 2) (yellow line:

12 month-olds, bright red line: 18-month-olds, orange line: 24-month-olds, brown line: 36-month-olds, turquoise line: primary school children, light blue line: adolescents, dark blue line: adults). **d** Group-wise violin plot of blink-per-trial ratio. Point represents mean estimate and error bars represent ± 1 SE. Distributions around the mean represent the density within the respective group. (yellow line: 12 month-olds ($n = 24$), bright red line: 18-month-olds ($n = 26$), orange line: 24-month-olds ($n = 30$), brown line: 36-month-olds ($n = 28$), turquoise line: primary school children ($n = 28$), light blue line: adolescents ($n = 30$), dark blue line: adults ($n = 64$)).

During the acquisition session on day 1, participants received 36 US–CS pairings in the same context. Additionally, two trials involving a puff of air alone were conducted at the start of the acquisition session, with the aim of testing participants' responsiveness to the air puff and aligning the tube correctly. Six blocks, each comprising eight trials with the following order, were conducted during the acquisition session: The trial order consisted of PPPPPAPT (P = paired trial, A = air puff alone and T = tone alone). Paired trials were conducted with a CS tone of 750 ms duration, overlapping and co-terminating with a 100 ms air puff directed towards the participant's left eye[43]. Therefore, the inter-stimulus interval was 650 ms. The inter-trial-interval randomly varied between 8 and 16 s. The session lasted about 12 min. During the second acquisition session on day 2, participants

received two additional blocks of acquisition trials (2 × PPPPPAPT) to consolidate their association learning[34,44].

## Procedure

A detailed description of the laboratory setup and procedure is available elsewhere[11]. Briefly, at the start of each session, there was a 10-min warm-up period to familiarise the infant with their surroundings and the experimenter. At the outset of the experiment, the experimenter attached the headband to the participant and adjusted the flexible tube. Infants sat on their parent's lap during testing. Parents could observe their child through a large mirror situated to the right hand side[10,11]. To maintain a straight line of sight with the eye-tracking camera, participants watched a 19-inch video

screen placed ~1 m ahead of them during each session. For the infants, the sessions involved the display of either Peppa Pig (Himmelsdrachen und andere Geschichten, 2003, Universal Pictures) or Timmy the Sheep (Timmy das Schäfchen spielt Fußball, 2011, Sony Music Entertainment). During the sessions, the experimenter would adjust the headband or air tube of the infants, if required, for which the experimenter sat on the right side of the 19-inch video screen facing the child. The experimenter had two hand puppets and building blocks kept nearby, in case the infant lost interest in the video being displayed. Primary school children, adolescents and adults sat on chairs in the testing room, facing a table with a laptop playing the video. Either 'The Planet' (Episode: Deserts, 2011, Polyband/WVG) or 'The Emperor's Journey' (2005, Bonne Pioche, Canal Plus, Buena Vista International) was shown to school-aged children, adolescents and adults.

### Data coding

Trained raters coded the video recordings of the sessions offline to identify eyeblink occurrences frame-by-frame using INTERACT software (Mangold International GmbH, Arnsdorf, Germany). Another rater independently coded acquisition session videos of 59 participants to examine inter-rater reliability. In sum, 3597 trials were rated independently. Inter-rater reliability was excellent, with an intra-class correlation coefficient of .897.

As in previous studies in young infants, blinks within 300 ms after tone onset were coded as startle or alpha responses[34]. A conditioned response was identified as an eyeblink occurring between 300 ms after tone onset and the air puff onset[34,43]. In the event that a conditioned response was observed, the onset time of this response was additionally coded as the interval between 300 ms after tone onset and the onset of eyelid closure. An unconditioned response was classified as an eyeblink occurring within 500 ms after air puff onset, on air puff only trials. A conditioned response during tone-alone trials was coded if eyeblinks were observed from 300 ms after tone onset until the unconditioned response period. This accounts for the potential difficulty some child age-groups may have with timing the blinking response[43,44].

### Statistical analyses

Data of the residuals of the LMM and the residuals of the piecewise LMM the data were tested for normal distribution using Q–Q-plots and histograms.

**Preregistered analyses.** Prior to the start of the study, the hypotheses were formulated with regard to the potential differences between infants, adolescents and adults. However, subsequent to the initiation of the study, it became evident that an important age-group between infancy and adolescents had been overlooked: that of primary school children. Consequently, an additional examination of this age cohort was conducted, although it is not included in the pre-registered analyses.

For the preregistered analyses, we were interested in how the amount of conditioned responses per block changed across session 1 and 2. The primary dependent variable to analyse the hypotheses was the percentage of conditioned responses (CRs) calculated per block of six paired trials (eight blocks in total). To analyse the hypotheses, a mixed-ANOVA with block (8) as a within-subject factor and age-group (12-, 18-, 24-, 36-months, adolescents, adults) as a between-subject factor on %CRs in paired trials was conducted. To analyse if there were significant increases in %CRs within each age-group, separate repeated-measures ANOVAs per age-group were conducted (H1a). Post-hoc t-tests were used to determine which blocks differed. To analyse differences between adults and infants, separate one-way ANOVAs with age-group as a between-subject factor were conducted on each block. Planned contrasts compared infants as a group (12–36 months) vs. adults (H1b.1) and then each infant age-group with each other (H1b.2). To analyse differences between adolescents and adults, post-hoc t-tests compared adolescents and adults with each other (H1c). To analyse gender differences, a mixed-ANOVA with age-group (12-, 18-, 24-, 36-months, adolescents, adults) and gender (female, male) as between-subject factors and block (8) as a within-subject factor on %CRs in paired trials was conducted.

**Table 1 | Daily learning rate by age-group as predicted by piecewise linear mixed model**

| Day | Age-group | Change | SE | CI$_{95\%}$ lower | CI$_{95\%}$ upper |
|---|---|---|---|---|---|
| Day 1 | 12 months | 31.49% | 2.74 | 26.11 | 36.86 |
| Day 1 | 18 months | 24.24% | 2.64 | 19.10 | 29.42 |
| Day 1 | 24 months | 11.71% | 2.45 | 6.91 | 16.51 |
| Day 1 | 36 months | 18.14% | 2.54 | 13.15 | 23.14 |
| Day 1 | Primary school children | 47.04% | 2.54 | 42.05 | 51.98 |
| Day 1 | Adolescents | 30.48% | 2.45 | 25.63 | 35.28 |
| Day 1 | Adults | 22.85% | 1.68 | 19.58 | 26.16 |
| Day 2 | 12 months | 48.71% | 2.45 | 43.91 | 53.51 |
| Day 2 | 18 months | 17.51% | 2.35 | 12.90 | 22.12 |
| Day 2 | 24 months | 39.47% | 2.18 | 35.17 | 43.75 |
| Day 2 | 36 months | 20.12% | 2.27 | 15.68 | 24.56 |
| Day 2 | Primary school children | 2.59% | 2.27 | −1.85 | 7.03 |
| Day 2 | Adolescents | 26.83% | 2.22 | 22.50 | 31.18 |
| Day 2 | Adults | 9.92% | 1.50 | 6.98 | 12.86 |

**Analyses of the rate of learning.** We sought to analyse the changes in the rate of learning using piecewise linear mixed modelling (LMM). This enables the evaluation of changes in conditioned responses over the course of the sessions. To evaluate changes in conditioned responses over time, we created a continuous measure reflecting time-varying blinking tendency using a sliding average over conditioned responses for each trial (blink yes/no) with a window size of 5 trials. Changes in blinking tendency were evaluated using a piecewise LMM that included two continuously coded time bins (day 1 and day 2), age-group (12-, 18-, 24-, 36-months, primary school children, adolescents, adults) and their cross-level interaction as fixed factors, with measurements nested within participants. Tests were considered significant at a two-sides significance level of alpha = 0.05. Coefficients were estimated with restricted maximum likelihood estimation using a Nelder-Mead optimiser. Significance of model factors was determined using Type III analysis of variance with Satterthwaite's method and effect sizes were reported as partial and its 95% confidence interval. Pairwise contrasts of age-specific trajectories were performed using the *emmeans::emtrends()* function with tukey-adjustment for multiple comparisons. These analyses were repeated for participants identified as 'learners' as identified by learning clusters.

**Analyses of the quantity of learning per day.** Next, the total numbers of conditioned responses per day (i.e., the amount of learning) were compared between groups using analogous LMM with a binary factor for day (day 1 vs. day 2), age-group and their interaction.

**Analyses of the onset of the conditioned response per day.** In a first step, the mean onset time of the conditioned responses per day was analysed per day using a one-way ANOVA. In a second step, a mixed-ANOVA was conducted with day (2) as a within-subject factor and age-group (12-, 18-, 24-, 36-months, primary school children, adolescents, adults) as a between-subject factor on the mean onset time of the eye closure for all participants who exhibited conditioned responses on both days.

**Analyses of learning cluster.** Distinct trajectories in learning rates over the course of the study were explored using time series clustering. Differences between each time series were calculated with the dynamic time warping method, which were then grouped using hierarchical cluster analysis. The number of clusters was selected using an adaptive branch

**Table 2 | Comparisons of daily learning rates between age-groups as predicted by piecewise linear mixed model**

| Day | Contrast | Difference | SE | z-value | P-value |
|-----|----------|-----------|-----|---------|---------|
| Day 1 | 12 months - 18 months | 7.25 | 3.79 | 1.90 | 0.4825 |
| Day 1 | 12 months - 24 months | 19.78 | 3.70 | 5.37 | <0.001*** |
| Day 1 | 12 months - 36 months | 13.34 | 3.74 | 3.56 | 0.00671** |
| Day 1 | 12 months - Primary school children | −15.55 | 3.74 | −4.15 | <0.001*** |
| Day 1 | 12 months - Adolescents | 1.01 | 3.70 | 0.28 | 0.99996 |
| Day 1 | 12 months - Adults | 8.59 | 3.22 | 2.67 | 0.10504 |
| Day 1 | 18 months - 24 months | 12.53 | 3.60 | 3.48 | 0.00906** |
| Day 1 | 18 months - 36 months | 6.10 | 3.65 | 1.67 | 0.63683 |
| Day 1 | 18 months - Primary school children | −22.75 | 3.65 | −6.21 | <0.001*** |
| Day 1 | 18 months - Adolescents | −6.19 | 3.60 | −1.72 | 0.60255 |
| Day 1 | 18 months - Adults | 1.39 | 3.12 | 0.44 | 0.99943 |
| Day 1 | 24 months - 36 months | −6.43 | 3.55 | −1.82 | 0.53521 |
| Day 1 | 24 months - Primary school children | −35.33 | 3.55 | −9.99 | <0.001*** |
| Day 1 | 24 months - Adolescents | −18.77 | 3.46 | −5.40 | <0.001*** |
| Day 1 | 24 months - Adults | −11.18 | 2.98 | −3.75 | 0.00336** |
| Day 1 | 36 months - Primary school children | −28.90 | 3.60 | −8.03 | <0.001*** |
| Day 1 | 36 months - Adolescents | −12.34 | 3.55 | −3.48 | 0.00896** |
| Day 1 | 36 months - Adults | −4.75 | 3.07 | −1.55 | 0.7132 |
| Day 1 | Primary school children - Adolescents | 16.56 | 3.55 | 4.68 | 6e-05 |
| Day 1 | Primary school children - Adults | 24.14 | 3.07 | 7.92 | <0.001*** |
| Day 1 | Adolescents - Adults | 7.58 | 2.98 | 2.55 | 0.14247 |
| Day 2 | 12 months - 18 months | 31.20 | 3.40 | 9.19 | <0.001*** |
| Day 2 | 12 months - 24 months | 9.24 | 3.29 | 2.82 | 0.07242 |
| Day 2 | 12 months - 36 months | 28.58 | 3.34 | 8.57 | <0.001*** |
| Day 2 | 12 months - Primary school children | 46.12 | 3.34 | 13.83 | <0.001*** |
| Day 2 | 12 months - Adolescents | 21.88 | 3.30 | 6.63 | <0.001*** |
| Day 2 | 12 months - Adults | 38.78 | 2.87 | 13.52 | <0.001*** |
| Day 2 | 18 months - 24 months | −21.96 | 3.22 | −6.84 | <0.001*** |
| Day 2 | 18 months - 36 months | −2.62 | 3.26 | −0.80 | 0.98491 |
| Day 2 | 18 months - Primary school children | 14.92 | 3.26 | 4.57 | 1e-04 |
| Day 2 | 18 months - Adolescents | −9.32 | 3.23 | −2.89 | 0.05938 |
| Day 2 | 18 months - Adults | 7.58 | 2.78 | 2.72 | 0.09298 |
| Day 2 | 24 months - 36 months | 19.34 | 3.14 | 6.14 | <0.001*** |
| Day 2 | 24 months - Primary school children | 36.88 | 3.14 | 11.71 | <0.001*** |
| Day 2 | 24 months - Adolescents | 12.62 | 3.12 | 4.05 | <0.001*** |
| Day 2 | 24 months - Adults | 29.54 | 2.65 | 11.14 | <0.001*** |
| Day 2 | 36 months - Primary school children | 17.53 | 3.20 | 5.47 | <0.001*** |
| Day 2 | 36 months - Adolescents | −6.71 | 3.17 | −2.12 | 0.34156 |
| Day 2 | 36 months - Adults | 10.20 | 2.71 | 3.76 | 0.00327** |
| Day 2 | Primary school children - Adolescents | −24.25 | 3.17 | −7.65 | <0.001*** |
| Day 2 | Primary school children - Adults | −7.33 | 2.71 | −2.70 | 0.09831 |
| Day 2 | Adolescents - Adults | 16.92 | 2.68 | 6.32 | <0.001*** |

*P* values were tukey-adjusted for multiple comparisons ** indicates *p* < 0.01, *** indicates *p* < 0.001.

**Table 3 | Mean blink-per-trial ratio displayed by day and age-group**

| Age-group | Day | Mean blink/trial ratio | SE | df | CI$_{95\%}$ lower | CI$_{95\%}$ upper |
|-----------|-----|------------------------|-----|-----|-------------------|-------------------|
| 12 months | Day 1 | 18.16% | 6.01 | 329.46 | 6.34 | 29.99 |
| 18 months | Day 1 | 11.67% | 5.78 | 329.46 | 0.31 | 23.03 |
| 24 months | Day 1 | 6.67% | 5.38 | 329.46 | −3.91 | 17.25 |
| 36 months | Day 1 | 14.33% | 5.57 | 329.46 | 3.38 | 25.27 |
| Primary school children | Day 1 | 56.03% | 5.57 | 329.46 | 45.08 | 66.98 |
| Adolescents | Day 1 | 27.77% | 5.38 | 329.46 | 17.19 | 38.35 |
| Adults | Day 1 | 33.12% | 3.68 | 329.46 | 25.88 | 40.36 |
| 12 months | Day 2 | 61.49% | 6.01 | 329.46 | 49.66 | 73.32 |
| 18 months | Day 2 | 32.35% | 5.78 | 329.46 | 20.99 | 43.71 |
| 24 months | Day 2 | 37.06% | 5.38 | 329.46 | 26.48 | 47.63 |
| 36 months | Day 2 | 34.82% | 5.57 | 329.46 | 23.87 | 45.77 |
| Primary school children | Day 2 | 71.73% | 5.57 | 329.46 | 60.78 | 82.67 |
| Adolescents | Day 2 | 56.65% | 5.44 | 336.96 | 45.96 | 67.35 |
| Adults | Day 2 | 48.7% | 3.68 | 329.46 | 41.46 | 55.94 |

pruning procedure for hierarchical clustering dendrograms (*dynamic-TreeCut::cutreeDynamic*) using the hybrid method with a minimum cluster size of 1[45]. Differences in cluster distributions between age-groups were evaluated using multinomial logistic regression. Significance was determined using $\chi^2$-likelihood ratio test.

### Reporting summary
Further information on research design is available in the Nature Portfolio Reporting Summary linked to this article.

## Results
### Preregistered analyses on the learning rate for blocks of 6 trials
Figure 1a presents the conditioned responses (blink/no blink) for every trial and participant, grouped by age. Note that only results on the preregistered age-groups are presented in this section, therefore leaving out the primary school children. The hypothesis that all age-groups will show a significant increase in CRs during the acquisition session was confirmed (H1a). A mixed-ANOVA revealed a significant main effect of *block*, $F_{(7, 1372)} = 82.70$, $p < 0.001$, $\eta p^2 = 0.297$ CI$_{95\%} = 0.26$–0.33, a significant main effect of *age-group*, $F_{(5, 196)} = 5.55$, $p < 0.001$, $\eta p^2 = 0.124$ CI$_{95\%} = 0.03$–0.19, and a significant interaction effect Block x Age-group, $F_{(35, 1372)} = 2.81$, $p < 0.001$, $\eta p^2 = 0.067$ CI$_{95\%} = 0.02$–0.07. To evaluate the source of this interaction, separate repeated-measures ANOVAs for each age-group were conducted. Results yielded significant increases in CRs in every age-group (12 months: $F_{(7,161)} = 26.11$, $p < 0.001$, $\eta p^2 = 0.53$ CI$_{95\%} = 0.41$–0.60; 18 months: $F_{(7,175)} = 7.56$, $p < 0.001$, $\eta p^2 = 0.23$ CI$_{95\%} = 0.11$–0.31; 24 months: $F_{(7,203)} = 15.99$, $p < 0.001$, $\eta p^2 = 0.36$ CI$_{95\%} = 0.23$–0.43; 36 months: $F_{(7,189)} = 10.47$, $p < 0.001$, $\eta p^2 = 0.28$ CI$_{95\%} = 0.15$–0.35; adolescents: $F_{(7,203)} = 12.96$, $p < 0.001$, $\eta p^2 = 0.31$ CI$_{95\%} = 0.18$–0.38; adults: $F_{(7,441)} = 15.84$, $p < 0.001$, $\eta p^2 = 0.20$ CI$_{95\%} = 0.13$–0.25). Post-hoc *t*-tests for each age-group revealed when the significant increase appeared. Recall that block 7 and 8 were administered during the second session. For adults, there was a significant increase in CRs between block 1–2, $M_{diff} = -8.28$, $t_{(63)} = -3.48$, $p = 0.001$, and between block 6–7, $M_{diff} = -10.94$, $t_{(63)} = -2.91$, $p = 0.005$. In adolescents, the significant increase occurred at the beginning of the second acquisition session: there was a significant increase from block 6–7, $M_{diff} = -27.11$, $t_{(29)} = -4.02$, $p < 0.001$. Furthermore, there was a significant decrease from block 7–8, $M_{diff} = 9.11$, $t_{(29)} = 2,13$, $p = 0.042$. For 12-month-olds, there were significant increases

**Table 4 | Daily age-group differences in blink-per-trial ratio**

| Contrast | Day | Difference | SE | df | t-value | P-value |
|---|---|---|---|---|---|---|
| 12 months - 18 months | Day 1 | 6.49% | 8.34 | 329.46 | 0.78 | 0.987 |
| 12 months - 24 months | Day 1 | 11.49% | 8.06 | 329.46 | 1.43 | 0.788 |
| 12 months - 36 months | Day 1 | 3.84% | 8.19 | 329.46 | 0.47 | 0.999 |
| 12 months - Primary school children | Day 1 | −37.86% | 8.19 | 329.46 | −4.62 | <0.001*** |
| 12 months - Adolescents | Day 1 | −9.61% | 8.06 | 329.46 | −1.19 | 0.897 |
| 12 months - Adults | Day 1 | −14.95% | 7.05 | 329.46 | −2.12 | 0.342 |
| 18 months - 24 months | Day 1 | 5.00% | 7.89 | 329.46 | 0.63 | 0.996 |
| 18 months - 36 months | Day 1 | −2.66% | 8.02 | 329.46 | −0.33 | 1 |
| 18 months - Primary school children | Day 1 | −44.36% | 8.02 | 329.46 | −5.53 | <0.001*** |
| 18 months - Adolescents | Day 1 | −16.10% | 7.89 | 329.46 | −2.04 | 0.391 |
| 18 months - Adults | Day 1 | −21.45% | 6.85 | 329.46 | −3.13 | 0.031* |
| 24 months - 36 months | Day 1 | −7.66% | 7.74 | 329.46 | −0.99 | 0.956 |
| 24 months - Primary school children | Day 1 | −49.36% | 7.74 | 329.46 | −6.38 | <0.001*** |
| 24 months - Adolescents | Day 1 | −21.10% | 7.60 | 329.46 | −2.78 | 0.084 |
| 24 months - Adults | Day 1 | −26.45% | 6.52 | 329.46 | −4.06 | 0.001** |
| 36 months - Primary school children | Day 1 | −41.70% | 7.87 | 329.46 | −5.30 | <0.001*** |
| 36 months - Adolescents | Day 1 | −13.44% | 7.74 | 329.46 | −1.74 | 0.591 |
| 36 months - Adults | Day 1 | −18.79% | 6.67 | 329.46 | −2.82 | 0.075 |
| Primary school children - Adolescents | Day 1 | 28.26% | 7.74 | 329.46 | 3.65 | 0.006** |
| Primary school children - Adults | Day 1 | 22.91% | 6.67 | 329.46 | 3.43 | 0.012* |
| Adolescents - Adults | Day 1 | −5.35% | 6.52 | 329.46 | −0.82 | 0.983 |
| 12 months - 18 months | Day 2 | 29.14% | 8.34 | 329.46 | 3.50 | 0.01* |
| 12 months - 24 months | Day 2 | 24.43% | 8.06 | 329.46 | 3.03 | 0.042* |
| 12 months - 36 months | Day 2 | 26.67% | 8.19 | 329.46 | 3.26 | 0.021* |
| 12 months - Primary school children | Day 2 | −10.24% | 8.19 | 329.46 | −1.25 | 0.874 |
| 12 months - Adolescents | Day 2 | 4.84% | 8.11 | 332.84 | 0.60 | 0.997 |
| 12 months - Adults | Day 2 | 12.79% | 7.05 | 329.46 | 1.81 | 0.539 |
| 18 months - 24 months | Day 2 | −4.70% | 7.89 | 329.46 | −0.60 | 0.997 |
| 18 months - 36 months | Day 2 | −2.47% | 8.02 | 329.46 | −0.31 | 1 |
| 18 months - Primary school children | Day 2 | −39.37% | 8.02 | 329.46 | −4.91 | <0.001*** |
| 18 months - Adolescents | Day 2 | −24.30% | 7.93 | 332.99 | −3.06 | 0.038* |
| 18 months - Adults | Day 2 | −16.35% | 6.85 | 329.46 | −2.39 | 0.208 |
| 24 months - 36 months | Day 2 | 2.23% | 7.74 | 329.46 | 0.29 | 1 |
| 24 months - Primary school children | Day 2 | −34.67% | 7.74 | 329.46 | −4.48 | <0.001*** |
| 24 months - Adolescents | Day 2 | −19.60% | 7.65 | 333.26 | −2.56 | 0.141 |
| 24 months - Adults | Day 2 | −11.64% | 6.52 | 329.46 | −1.79 | 0.558 |
| 36 months - Primary school children | Day 2 | −36.91% | 7.87 | 329.46 | −4.69 | <0.001*** |
| 36 months - Adolescents | Day 2 | −21.83% | 7.78 | 333.13 | −2.81 | 0.077 |
| 36 months - Adults | Day 2 | −13.88% | 6.67 | 329.46 | −2.08 | 0.367 |
| Primary school children - Adolescents | Day 2 | 15.07% | 7.78 | 333.13 | 1.94 | 0.457 |
| Primary school children - Adults | Day 2 | 23.03% | 6.67 | 329.46 | 3.45 | 0.011* |
| Adolescents - Adults | Day 2 | 7.95% | 6.57 | 334.61 | 1.21 | 0.89 |

P values were tukey-adjusted for multiple comparisons * indicates $p < 0.05$, ** indicates $p < 0.01$, *** indicates $p < 0.001$.

between block 2 and block 3, $M_{diff} = -9.75$, $t_{(23)} = -2.21$, $p = 0.041$, between block 6 and 7, $M_{diff} = -26.67$, $t_{(23)} = -4.05$, $p = 0.001$, and between block 7 and 8, $M_{diff} = -17.36$, $t_{(23)} = -3.67$, $p = 0.001$. For 18-month-olds, there was a significant increase between block 6 and 7, $M_{diff} = -17.69$, $t_{(25)} = -3.11$, $p = 0.005$. For 24-month-olds, there were significant increases between block 1 and 2, $M_{diff} = -5.00$, $t_{(29)} = -2.19$, $p = 0.037$, between block 6 and 7, $M_{diff} = -17.45$, $t_{(29)} = -2.90$, $p = 0.007$, and block 7 and 8, $M_{diff} = -14.78$, $t_{(29)} = -3.16$, $p = 0.004$. For 36-month-olds, there were significant increases between block 1 and 2, $M_{diff} = -5.95$, $t_{(27)} = -2.26$, $p = 0.032$, between block

2 and 3, $M_{diff} = -7.98$, $t_{(27)} = -2.18$, $p = 0.039$, and block 7 and 8, $M_{diff} = -11.43$, $t_{(27)} = -2.16$, $p = 0.040$.

Furthermore, the hypothesis that adults will acquire the association faster than infants was confirmed (H1b.1). Separate one-way ANOVAs for each block revealed that there were significant differences between age-groups in each block (block 1: $F_{(5)} = 6.67$, $p < 0.001$, $\eta p^2 = 0.145$ $CI_{95\%} = 0.05-0.22$; block 2: $F_{(5)} = 6.09$, $p < 0.001$, $\eta p^2 = 0.134$ $CI_{95\%} = 0.04-0.21$; block 3: $F_{(5)} = 4.40$, $p < 0.001$, $\eta p^2 = 0.101$ $CI_{95\%} = 0.02-0.17$; block 4: $F_{(5)} = 5.91$, $p < 0.001$, $\eta p^2 = 0.131$

**Table 5 | Within-group differences in blink-per-trial ratio day 1 vs. day 2**

| Contrast | Age-group | Difference | SE | df | *t*-value | *P*-value |
|---|---|---|---|---|---|---|
| Day 1 - Day 2 | 12 months | −43.33% | 5.42 | 222.04 | −7.99 | <0.001*** |
| Day 1 - Day 2 | 18 months | −20.68% | 5.21 | 222.04 | −3.97 | <0.001*** |
| Day 1 - Day 2 | 24 months | −30.39% | 4.85 | 222.04 | −6.26 | <0.001*** |
| Day 1 - Day 2 | 36 months | −20.50% | 5.02 | 222.04 | −4.08 | <0.001*** |
| Day 1 - Day 2 | Primary school children | −15.70% | 5.02 | 222.04 | −3.13 | 0.002** |
| Day 1 - Day 2 | Adolescents | −28.88% | 4.92 | 224.47 | −5.87 | <0.001*** |
| Day 1 - Day 2 | Adults | −15.58% | 3.32 | 222.04 | −4.69 | <0.001*** |

*P* values were tukey-adjusted for multiple comparisons ** indicates *p* < 0.01, *** indicates *p* < 0.001.

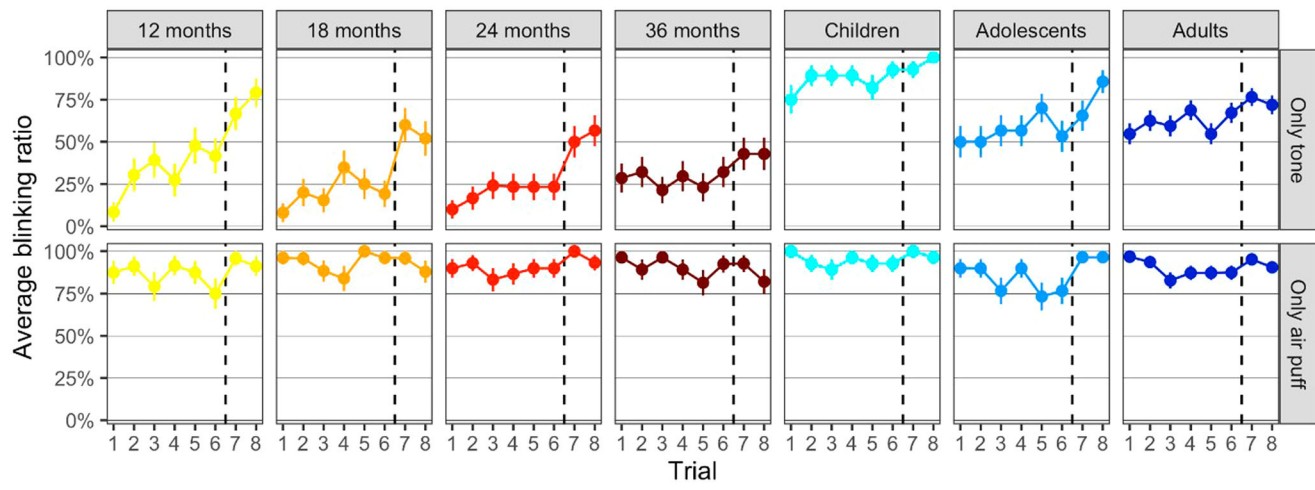

**Fig. 2 | Mean blinking tendencies per tone alone and air puff alone trials for each age-group.** N per age-group (12 month-olds: *n* = 24, 18-month-olds: *n* = 26, 24-month-olds: *n* = 30, 36-month-olds: *n* = 28, primary school children: *n* = 28, adolescents: *n* = 30, adults: *n* = 64). Trials 1–6 were administered on day 1, and trial 7–8 were administered on day 2. Dashed lines mark the transition from day 1 to day 2.

**Table 6 | Age-group differences in conditioned responses on tone alone trials**

| Contrast | Mean difference | *t*-value | SE | *p*-value |
|---|---|---|---|---|
| Primary school children - 12 months | 50.83 | 5.94 | 8.56 | <0.001 |
| Primary school children - 18 months | 64.50 | 7.70 | 8.38 | <0.001 |
| Primary school children - 24 months | 63.07 | 7.80 | 8.09 | <0.001 |
| Primary school children - 36 months | 58.32 | 7.09 | 8.23 | <0.001 |
| Primary school children - Adolescents | 26.47 | 3.27 | 8.09 | 0.001 |
| Primary school children - Adults | 24.61 | 3.53 | 6.97 | <0.001 |
| Adolescents - 12 months | 24.37 | 2.89 | 8.43 | 0.004 |
| Adolescents - 18 months | 38.03 | 4.61 | 8.25 | <0.001 |
| Adolescents - 24 months | 36.60 | 4.60 | 7.95 | <0.001 |
| Adolescents - 36 months | 31.86 | 3.94 | 8.09 | <0.001 |
| Adolescents - adults | −1.86 | −0.27 | 6.81 | 0.785 |
| Adults - 12 months | 26.22 | 3.56 | 7.37 | <0.001 |
| Adults - 18 months | 39.89 | 5.57 | 7.16 | <0.001 |
| Adults - 24 months | 38.46 | 5.65 | 6.81 | <0.001 |
| Adults - 36 months | 33.71 | 4.84 | 6.97 | <0.001 |

$CI_{95\%} = 0.04$–$0.20$; block 5: $F_{(5)} = 4.22$, $p < 0.001$, $\eta p^2 = 0.097$ $CI_{95\%} = 0.02$–$0.16$; block 6: $F_{(5)} = 4.70$, $p < 0.001$, $\eta p^2 = 0.107$ $CI_{95\%} = 0.02$–$0.17$; block 7: $F_{(5)} = 4.53$, $p < 0.001$, $\eta p^2 = 0.104$ $CI_{95\%} = 0.02$–$0.17$; block 8: $F_{(5)} = 2.79$, $p = 0.019$, $\eta p^2 = 0.066$ $CI_{95\%} = 0.00$–$0.12$). Planned contrasts revealed higher %CRs in adults compared to all infant age-groups as a group in block 1–7 (block 1: $t_{(196)} = 5.37$, $p < 0.001$; block 2: $t_{(196)} = 5.37$, $p < 0.001$; block 3: $t_{(196)} = 4.15$, $p < 0.001$; block 4: $t_{(196)} = 4.58$, $p < 0.001$; block 5: $t_{(196)} = 3.85$, $p < 0.001$; block 6: $t_{(196)} = 4.24$, $p < 0.001$; block 7: $t_{(196)} = 2.53$, $p = 0.012$). There was no statistically significant difference in block 8, $t_{(196)} = 0.66$, $p = 0.947$,.

The hypothesis that acquisition rates are higher, the older the infant was rejected (H1b.2). Planned contrasts revealed that 12-month-olds had significant higher %CRs than 24-month-olds in block 4 ($t_{(196)} = 2.19$, $p = 0.030$), and than 18-, 24-, 36-month-olds in block 7 ($t_{(196)} = 2.07$, $p = 0.039$; $t_{(196)} = 2.42$, $p = 0.017$, $t_{(196)} = 2.39$, $p = 0.018$, respectively) and 8 ($t_{(196)} = 3.48$, $p < 0.001$; $t_{(196)} = 2.50$, $p < 0.001$; $t_{(196)} = 2.79$, $p < 0.001$, respectively). Thus, contrary to the hypothesis, 12-month-olds seemed to be the fastest learners within the infant sample.

Lastly, the hypothesis that adults will acquire the association between the tone and the air puff faster than adolescents was rejected (H1c). Post-hoc *t*-tests revealed no statistically significant differences between adults and adolescents in any of the blocks (block 1: $t_{(92)} = -7.34$, $p = 0.232$, $d = -0.162$ $CI_{95\%} = -0.60$ to $0.27$; block 2: $t_{(82.287)} = -1.60$, $p = 0.056$, $d = -0.306$ $CI_{95\%} = -0.74$ to $0.13$; block 3: $t_{(92)} = -0.624$, $p = 0.267$, $d = -0.138$ $CI_{95\%} = -0.57$ to $0.30$; block 4: $t_{(92)} = -0.620$, $p = 0.268$, $d = -0.137$ $CI_{95\%} = -0.57$ to $0.30$; block 5: $t_{(92)} = -0.309$, $p = 0.349$, $d = -0.086$ $CI_{95\%} = -0.52$ to $0.35$; block 6: $t_{(92)} = -0.704$, $p = 0.242$, $d = -0.156$

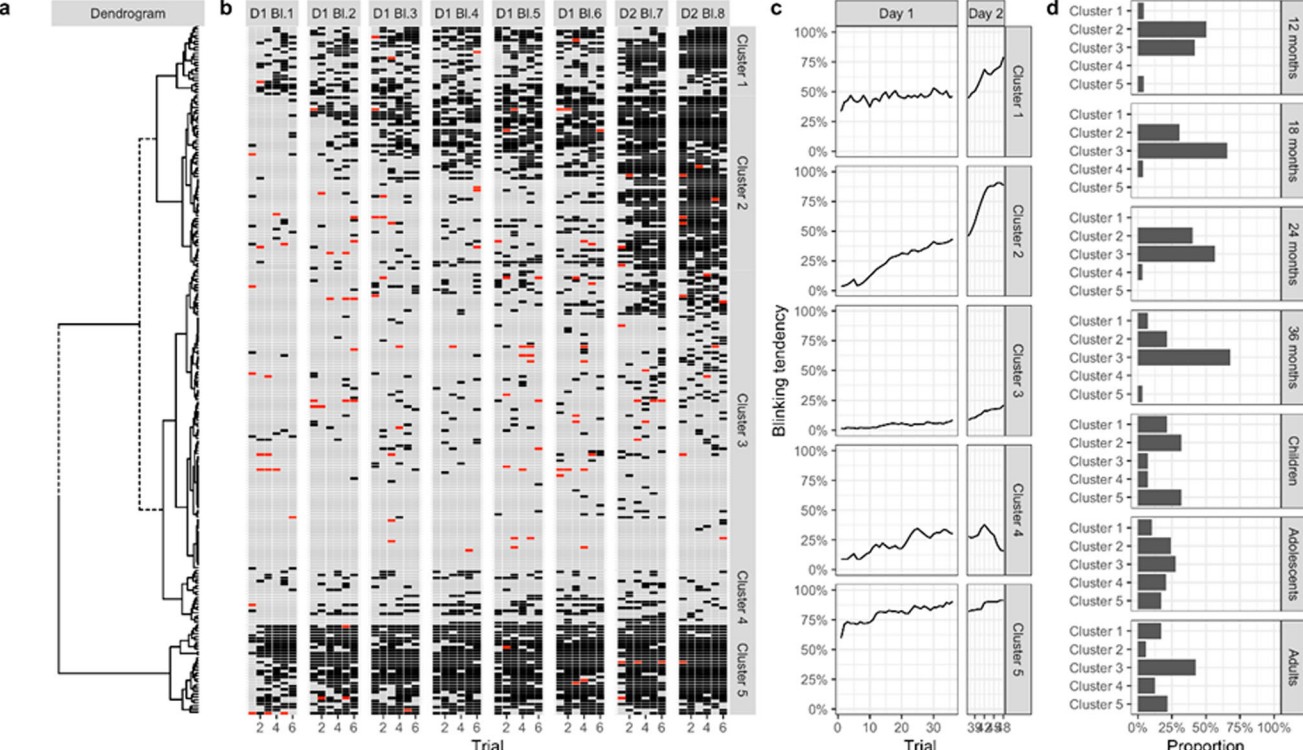

**Fig. 3 | Learning Cluster and their distribution across age-groups (12 month-olds: $n = 24$, 18-month-olds: $n = 26$, 24-month-olds: $n = 30$, 36-month-olds: $n = 28$, primary school children: $n = 28$, adolescents: $n = 30$, adults: $n = 64$). a** Dendrogram of the hierarchical cluster analysis on conditioned responses.

**b** Conditioned responses for each trial and each participant in each cluster (black squares: blink yes/grey squares: no blink/red squares: missing). **c** Blinking trajectories over the course of day 1 and 2 per learning cluster. **d** Distribution of learning clusters across age-groups.

**Table 7 | Daily learning rate by cluster as predicted by piecewise linear mixed model**

| Day | Cluster | Change | SE | CI$_{95\%}$ lower | CI$_{95\%}$ upper |
|---|---|---|---|---|---|
| Day 1 | Cluster 1 | 8.88% | 2.54 | 3.89 | 13.87 |
| Day 1 | Cluster 2 | 60.10% | 1.58 | 56.93 | 63.22 |
| Day 1 | Cluster 3 | 8.45% | 1.20 | 6.10 | 10.85 |
| Day 1 | Cluster 4 | 36.10% | 2.88 | 30.48 | 41.71 |
| Day 1 | Cluster 5 | 24.00% | 2.21 | 19.63 | 28.37 |
| Day 2 | Cluster 1 | 27.08% | 2.27 | 22.64 | 31.52 |
| Day 2 | Cluster 2 | 52.70% | 1.43 | 49.91 | 55.50 |
| Day 2 | Cluster 3 | 13.72% | 1.08 | 11.59 | 15.84 |
| Day 2 | Cluster 4 | −13.04% | 2.56 | −18.06 | −8.03 |
| Day 2 | Cluster 5 | 1.12% | 1.98 | −2.77 | 5.00 |

$CI_{95\%} = -0.59$ to $0.28$; block 7: $t_{(92)} = 1.456$, $p = 0.074$, $d = 0.322$ $CI_{95\%} = -0.12$ to $0.76$; block 8: $t_{(92)} = 0.498$, $p = 0.310$, $d = 0.109$ $CI_{95\%} = -0.32$ to $0.54$).

To examine gender differences in acquisition of the CR, a mixed ANOVA with age-group and gender as between-subject factors and block as a within-subject factor was conducted. There was no evidence for a main effect of gender, $F_{(1, 190)} = 0.59$, $p = 0.445$, $\eta^2 = 0.003$ $CI_{95\%} = 0.00–0.04$ and no evidence for an interaction effect between age-group and gender, $F_{(5, 190)} = 0.48$, $p = 0.793$, $\eta p^2 = 0.012$ $CI_{95\%} = 0.00–0.03$.

**Age differences in the rate of learning on a trial-by-trial basis per day.** Additionally, we conducted a comparison of the increase in conditioned responses over time on a trial-by-trial basis per age-group. Figure 1b

displays the mean conditioned responses for each age-group over the course of day 1 and 2. There was a significant increase in conditioned responses over the course of the first ($F_{(1,10788)} = 815.90$, $p < 0.001$, $\eta_P^2 = 0.07$ $CI_{95\%} = 0.05–0.08$) and the second acquisition day ($F_{(1,10788.20)} = 808.78$, $p < 0.001$, $\eta_P^2 = 0.07$ $CI_{95\%} = 0.05–0.08$). Age-groups differed in their learning rates on day 1 ($F_{(6, 10788)} = 20.47$, $p < .001$, $\eta_P^2 = 0.01$ $CI_{95\%} = 0.01–0.02$) and day 2 ($F_{(6, 10788.17)} = 54.89$, $p < 0.001$, $\eta_P^2 = 0.03$ $CI_{95\%} = 0.02–0.04$). During day 1, there were significant increases in conditioned responses in all age-groups (see Table 1). During day 2, all age-groups except for primary school children also showed a significant increase in conditioned responses. Primary school children reached their learning asymptote on day 1 and maintained the high level of conditioned responses on day 2.

Next, we analysed these differences using post-hoc tests (see Table 2). Comparing the learning rates of each age-group on day 1 and day 2, again there was no evidence for an increase in learning speed from 12 to 36 months, contrary to the hypothesis (see Fig. 1b). Twelve-month-olds showed the highest conditioned responses and steepest learning curves of the infant sample. Primary school children had a significantly steeper learning curve than all other age-groups on day 1 (see Table 2). Adolescents had a steeper learning curve on day 2 than adults (see Table 2). Adults had a steeper learning curve than 24-month-olds on day 1, and a steeper learning curve than 24- and 36-month-olds on day 2. On day 2, all age-groups except for adults had steeper learning curves than primary school children, as the latter did not make any additional learning gains. While infant age-groups showed relatively flat learning curves on day 1, there was a steep increase from day 1 to day 2. Thus, infants seem to rely heavily on a second acquisition session in order to show successful learning.

**Age differences in the quantity of learning per day.** Next, we compared the quantity of learning between age-groups per day (i.e., *how many* conditioned responses occurred per day relative to the number of

**Table 8 | Learning rate cluster distributions within age-groups**

| Cluster | 12 months | 18 months | 24 months | 36 months | Primary school children | Adolescents | Adults |
|---|---|---|---|---|---|---|---|
| Cluster 1 | 1 (4.17) | 0 (0) | 0 (0) | 2 (7.14) | 6 (21.43) | 3 (10.34) | 11 (17.19) |
| Cluster 2 | 12 (50) | 8 (30.77) | 12 (40) | 6 (21.43) | 9 (32.14) | 7 (24.14) | 4 (6.25) |
| Cluster 3 | 10 (41.67) | 17 (65.38) | 17 (56.67) | 19 (67.86) | 2 (7.14) | 8 (27.59) | 27 (42.19) |
| Cluster 4 | 0 (0) | 1 (3.85) | 1 (3.33) | 0 (0) | 2 (7.14) | 6 (20.69) | 8 (12.5) |
| Cluster 5 | 1 (4.17) | 0 (0) | 0 (0) | 1 (3.57) | 9 (32.14) | 5 (17.24) | 14 (21.88) |

Values represent $n$ (%), Cluster assignments were determined using time series clustering with dynamic time warping.

trials), indicated by the total amount of conditioned responses on day 1 and 2 (see Table 3). Percent of conditioned responses per age-group and day are displayed in Fig. 1d. The quantity of learning increased significantly from day 1 to day 2 ($F_{(1, 222.53)} = 184.93$, $p < 0.001$, $\eta_p^2 = 0.45$ $CI_{95\%} = 0.36–0.53$). There was a significant interaction between time and age-group ($F_{(6, 222.53)} = 4.22$, $p < 0.001$, $\eta_p^2 = 0.10$ $CI_{95\%} = 0.02–0.16$). On day 1, primary school children blinked significantly more than any other age-group (all tukey-adjusted $ps < 0.05$) and adults blinked significantly more than 18- and 24-month-olds (see Table 4). On day 2, both primary school children and 12-month-olds blinked more than 18-, 24- and 36-month-olds, primary school children blinked more than adults, and adolescents more than 18-month-olds (see Table 4). Comparing the performance across sessions, all age-groups showed a significantly higher number of conditioned responses on day 2 compared to day 1 with the highest increase in 12-month-olds (=43%) and the lowest in adults and primary school children (both = 16%) (see Table 5).

As an additional measure of learning, the quantity of conditioned responses to 8 tone alone trials that were administered after every sixth paired trials was examined. Tone alone trials represent a test for successful association learning. Figure 2 shows the blink ratios for each of the tone alone trials for each age-group. The age-groups differed significantly in how many tone alone trials they showed a conditioned response (Welch's $F_{(6, 87.80)} = 29.73$, $p < 0.001$, $\eta_p^2 = 0.34$ $CI_{95\%} = 0.23–0.41$). Post-hoc tests indicated that adults, adolescents and primary school-aged children showed more conditioned responses on tone alone trials than each infant age-group, and that primary school-aged children showed more conditioned responses on tone alone trials than adolescents and adults (see Table 6).

The age-groups did not exhibit statistically significant differences in their responsiveness to the air puff ($F_{(6, 229)} = 1.07$, $p = 0.379$, $\eta_p^2 = 0.28$ $CI_{95\%} = 0.00–0.06$), as can be seen in Fig. 2, bottom panel.

**Age differences in the mean onset time of the conditioned response per day.** Mean onset time of the conditioned response on day 1 was 507 ms ($SD = 28$ ms) for 12-month-olds, 511 ms ($SD = 87$ ms) for 18-month-olds, 533 ms ($SD = 79$ ms) for 24-month-olds, 490 ms ($SD = 78$ ms) for 36-month-olds, 508 ms ($SD = 39$ ms) for primary school children, 512 ms ($SD = 64$ ms) for adolescents and 519 ms ($SD = 83$ ms) for adults. Mean onset time of the conditioned response on day 2 was 512 ms ($SD = 77$ ms) for 12-month-olds, 464 ms ($SD = 74$ ms) for 18-month-olds, 473 ms ($SD = 54$ ms) for 24-month-olds, 483 ms ($SD = 84$ ms) for 36-month-olds, 470 ms ($SD = 61$ ms) for primary school children, 511 ms ($SD = 66$ ms) for adolescents and 495 ms ($SD = 75$ ms) for adults. Mean onset time of the conditioned responses did not differ statistically significantly between age-groups on day 1 ($F_{(6, 183)} = 0.74$, $p = 0.631$, $\eta_p^2 = 0.024$ $CI_{95\%} = 0.00–0.05$), or day 2 ($F_{(6, 193)} = 1.69$, $p = 0.125$, $\eta_p^2 = 0.050$ $CI_{95\%} = 0.00–0.09$). A mixed-ANOVA with day (1, 2) as a within-subject factor and age-group (12-, 18-, 24-, 36-months, primary school children, adolescents, adults) as a between-subject factor on the onset time of the eye closure for all participants who exhibited conditioned responses on both days revealed an effect of day ($F_{(1, 166)} = 11.20$, $p = 0.001$, $\eta_p^2 = 0.063$ $CI_{95\%} = 0.01–0.14$), but no evidence of an effect of age-group ($F_{(6, 166)} = 0.77$, $p = 0.599$, $\eta_p^2 = 0.027$ $CI_{95\%} = 0.00–0.05$) and

no evidence of an interaction effect ($F_{(6, 166)} = 1.00$, $p = 0.428$, $\eta_p^2 = 0.035$ $CI_{95\%} = 0.00–0.07$). A post-hoc paired t-test showed that the conditioned responses occurred earlier (i.e., shorter latencies) on day 2 than on day 1, $t_{(172)} = 3.55$, $p < 0.001$, $d = 0.270$ $CI_{95\%} = 0.12–0.42$.

**Learning cluster.** The individual learning curves per age-group showed that there was considerable variance between and within each age-group (see Fig. 1d). Next, we performed a cluster analysis to identify different learning trajectories. The learning clusters are displayed in Fig. 3a–c. As visible, five different learning clusters emerged (see Table 7). Cluster 1 represented a relatively high entry level and a high and stable level of conditioned responses with an increase on the 2nd day. Cluster 2 was the second most common cluster and described a very steep learning curve resulting in a high level of conditioned responses. Cluster 3 was the most common cluster and contained the non-learners who did not show a considerable increase in conditioned responses. Cluster 4 described varying levels of conditioned responses but with no meaningful increase across sessions and a decrease during day 2. Cluster 5 was a learning curve showing immediate learning that remained high across the blocks. Figure 3d shows the distribution of learning clusters across age-groups (also see Table 8).

Infants demonstrated either a rapid learning curve, with significant improvements from day 1 to day 2 (Cluster 2), or no progress at all (Cluster 3). In contrast, primary school-aged children exhibited either immediate learning (Clusters 1 and 5), or a rapid increase in learning (Cluster 2). Adolescents showed the most diverse distribution of learning types, with nearly equal distributions across all learning clusters. In adults, however, more than half of the participants demonstrated either a lack of learning (Clusters 3 and 4), or fast-paced learning (Clusters 1 and 5).

**Analysis of the learning rate of learners only.** Next, we analysed the learning rates of each age-group, solely based on the learners identified in the cluster analyses, to ascertain the veracity of our initial findings. Only participants from Clusters 1, 2 and 5 were included in the analyses. On Day 1, infant learners from all infant age-groups with the exception of 24-month-olds exhibited greater increases than adults (see Table 9). Primary school children demonstrated larger increases than adults, and adolescents exhibited larger increases than adults (Table 9). On day 2, infant age-groups exhibited the most substantial increases in conditioned responses, with gains ranging from 47 to 87% (see Table 10). In contrast, primary school children and adult learners had already reached their learning asymptote by day 1 and demonstrated only marginal learning increases on day 2 (Table 10). Adolescent learners also demonstrated medium increases of 36% from the second acquisition session, indicating that they benefited from the additional learning opportunity.

## Discussion

This is the first study to investigate associative learning from infancy to adulthood using the same paradigm. We found that all age-groups, from 12-month-old infants to adults, successfully learned the association between the

**Table 9 | Comparisons of the daily learning rates of learners between age-groups as predicted by piecewise linear mixed model**

| Period | Contrast | Difference | SE | z-value | P-value |
|---|---|---|---|---|---|
| Day 1 | 12 months - 18 months | −8.88 | 6.53 | −1.37 | 0.8201 |
| Day 1 | 12 months - 24 months | 23.81 | 5.76 | 4.12 | <0.001*** |
| Day 1 | 12 months - 36 months | 3.74 | 6.29 | 0.60 | 0.99695 |
| Day 1 | 12 months - Children | −0.14 | 4.94 | −0.03 | 1 |
| Day 1 | 12 months - Adolescents | 9.50 | 5.47 | 1.74 | 0.58873 |
| Day 1 | 12 months - Adults | 33.31 | 4.80 | 6.97 | <0.001*** |
| Day 1 | 18 months - 24 months | 32.74 | 6.72 | 4.88 | <0.001*** |
| Day 1 | 18 months - 36 months | 12.62 | 7.15 | 1.77 | 0.56869 |
| Day 1 | 18 months - Children | 8.74 | 6.00 | 1.46 | 0.77008 |
| Day 1 | 18 months - Adolescents | 18.38 | 6.43 | 2.86 | 0.06418 |
| Day 1 | 18 months - Adults | 42.24 | 5.86 | 7.20 | <0.001*** |
| Day 1 | 24 months - 36 months | −20.11 | 6.48 | −3.10 | 0.03174* |
| Day 1 | 24 months - Children | −24.00 | 5.18 | −4.62 | <0.001*** |
| Day 1 | 24 months - Adolescents | −14.35 | 5.71 | −2.52 | 0.15262 |
| Day 1 | 24 months - Adults | 9.50 | 5.04 | 1.88 | 0.49092 |
| Day 1 | 36 months - Children | −3.89 | 5.76 | −0.68 | 0.99382 |
| Day 1 | 36 months - Adolescents | 5.76 | 6.19 | 0.93 | 0.96782 |
| Day 1 | 36 months - Adults | 29.62 | 5.62 | 5.28 | <0.001*** |
| Day 1 | Children - Adolescents | 9.65 | 4.85 | 2.00 | 0.41696 |
| Day 1 | Children - Adults | 33.50 | 4.03 | 8.26 | <0.001*** |
| Day 1 | Adolescents - Adults | 23.81 | 4.66 | 5.10 | <0.001*** |
| Day 2 | 12 months - 18 months | 14.23 | 5.80 | 2.45 | 0.17639 |
| Day 2 | 12 months - 24 months | −26.80 | 5.15 | −5.21 | <0.001*** |
| Day 2 | 12 months - 36 months | 14.32 | 5.59 | 2.56 | 0.13845 |
| Day 2 | 12 months - Children | 58.54 | 4.40 | 13.30 | <0.001*** |
| Day 2 | 12 months - Adolescents | 24.26 | 4.86 | 4.99 | <0.001*** |
| Day 2 | 12 months - Adults | 45.35 | 4.26 | 10.65 | <0.001*** |
| Day 2 | 18 months - 24 months | −41.03 | 5.98 | −6.87 | <0.001*** |
| Day 2 | 18 months - 36 months | 0.08 | 6.36 | 0.01 | 1 |
| Day 2 | 18 months - Children | 44.30 | 5.34 | 8.29 | <0.001*** |
| Day 2 | 18 months - Adolescents | 10.03 | 5.72 | 1.75 | 0.58183 |
| Day 2 | 18 months - Adults | 31.12 | 5.23 | 5.95 | <0.001*** |
| Day 2 | 24 months - 36 months | 41.11 | 5.77 | 7.12 | <0.001*** |
| Day 2 | 24 months - Children | 85.33 | 4.63 | 18.44 | <0.001*** |
| Day 2 | 24 months - Adolescents | 51.06 | 5.06 | 10.07 | <0.001*** |
| Day 2 | 24 months - Adults | 72.14 | 4.49 | 16.06 | <0.001*** |
| Day 2 | 36 months - Children | 44.22 | 5.11 | 8.65 | <0.001*** |
| Day 2 | 36 months - Adolescents | 9.95 | 5.52 | 1.80 | 0.54594 |
| Day 2 | 36 months - Adults | 31.03 | 4.99 | 6.21 | <0.001*** |
| Day 2 | Children - Adolescents | −34.27 | 4.31 | −7.96 | <0.001*** |
| Day 2 | Children - Adults | −13.19 | 3.61 | −3.65 | 0.00484** |
| Day 2 | Adolescents - Adults | 21.08 | 4.16 | 5.07 | <0.001*** |

P values were tukey-adjusted for multiple comparisons * indicates $p < 0.05$, ** indicates $p < 0.01$, *** indicates $p < 0.001$.

**Table 10 | Daily learning rate of learners per age-group as predicted by piecewise linear mixed model**

| Day | Age-group | Change | SE | CI$_{95\%}$ lower | CI$_{95\%}$ upper |
|---|---|---|---|---|---|
| Day 1 | 12 months | 51.94% | 3.94 | 44.21 | 59.62 |
| Day 1 | 18 months | 60.82% | 5.18 | 50.64 | 70.99 |
| Day 1 | 24 months | 28.08% | 4.22 | 19.78 | 36.38 |
| Day 1 | 36 months | 48.19% | 4.90 | 38.59 | 57.79 |
| Day 1 | Children | 52.08% | 2.98 | 46.18 | 57.94 |
| Day 1 | Adolescents | 42.43% | 3.79 | 34.99 | 49.82 |
| Day 1 | Adults | 18.58% | 2.74 | 13.25 | 23.90 |
| Day 2 | 12 months | 60.53% | 3.49 | 53.66 | 67.38 |
| Day 2 | 18 months | 46.30% | 4.63 | 37.22 | 55.36 |
| Day 2 | 24 months | 87.32% | 3.78 | 79.92 | 94.73 |
| Day 2 | 36 months | 46.21% | 4.37 | 37.66 | 54.76 |
| Day 2 | Children | 1.99% | 2.68 | −3.25 | 7.22 |
| Day 2 | Adolescents | 36.26% | 3.38 | 29.64 | 42.89 |
| Day 2 | Adults | 15.18% | 2.44 | 10.42 | 19.94 |

hypothesis of age-related improvements in implicit memory development. Instead, they suggest the possibility of an approximate inverted U-shaped learning curve. Primary school children performed better than any other age-group, displaying the most consistent and least variable learning. Associative learning appears to be particularly important and adaptive in primary school-aged children[4]. In preschool and early school years, children may have an advantage in associative learning due to an increase in conscious, controlled cognitive processes related to contingency awareness, enabling them to proactively anticipate events[46,47]. This could help them achieve their developmental goals of gaining personal independence and quickly develop mental skills necessary during the early school years[48]. In line with this, recent studies indicate that primary school children can learn more items within a given period of time than adults[49] and demonstrate the most significant improvement in performance on an implicit skill task[25].

Age-related differences during associative learning were most prominent at the beginning of the acquisition session. During the first session, both primary school-aged children and adults displayed rapid learning. Adults exhibited a steeper learning curve and produced more conditioned responses than infants did. Adults showed a reduced rate of learning during the second session. This suggests a form of rapid learning, which may be effective enough for adults to comprehend the paradigm and thus triggering quick habituation.

Infants between 12 and 36 months of age required a follow-up session to demonstrate successful learning, in line with previous studies of 4–5-month-olds[34,36]. In infants, there were minor increases within the first trials, but the largest rise in conditioned responses occurred at the start of session 2. Therefore, consolidation, perhaps preferentially occurring during sleep, may be essential for younger age-groups to retain information[50,51]. There was no evidence for age-related increase of learning speed within our infant sample. In fact, 12-month-olds were the fastest learners in our study. However, additional analyses revealed that this result was only significant due to unequal distribution of learners and non-learners within the infant age-groups. Our results show that, in addition to the mean values of the conditional responses for the entire sample, additional finer grained analyses are required to understand changes associative learning.

Further analyses were conducted on the learning rates of the learners only. The results obtained from the primary school children, adolescents and adults were found to be consistent. However, the finding that infants in the learner group exhibited a steeper learning curve than adults is particularly intriguing. One possible interpretation of this is that infants who successfully acquire the association may have exhibited a stronger initial response to the air puff, which could have resulted in a steeper initial learning curve. This may be attributed to heightened sensitivity to novelty or

tone and the air puff, as evidenced by an increase in conditioned responses over the course of the experiment. Furthermore, our findings show that associative learning undergoes significant changes across development. Although earlier behavioural data suggested a linear improvement in speed from infancy to adulthood, our data indicate a peak at primary school age. The results do not support the invariant hypothesis, nor do they support the

**Article**

a greater focus on establishing new associations during this critical developmental period. The reason for the significant discrepancy in the number of learners and non-learners remains a topic for further investigation.

In contrast to the study by Löwgren et al.[38] which found gender differences in 6–11-year-old children and adults, there was no evidence for gender differences in acquisition in the present sample. As the authors mentioned, there were only nine male adult participants in their study and thus caution is warranted in interpreting the data. In line with our finding, a recent study by Vieites et al.[52] found no evidence for gender differences in 3–6-year-old children using trace eyeblink conditioning. In animals, sex differences in learning in eyeblink conditioning emerged only after puberty, whereas no evidence for sex differences were observed before or during puberty[41], which partially explains our results of missing gender differences in children and adolescents. Since gender differences can be mostly explained by hormone levels, future studies in adults using eyeblink conditioning should take into account female cycling levels and hormonal contraception[41].

Learning rates during acquisition varied considerably among participants. Different learning clusters indicated that there were numerous participants, including about half of the adult sample, who failed to acquire the association. Adolescent learners exhibited the most remarkable variability. Although they did not differ in the quantity of conditioned responses to adults, they had steeper learning curves than adults. It has to be noted that our adolescent sample had a wide age range from 12 to 17 years, possibly masking age-related changes in learning. It is possible that subgroups within the adolescent age range may be overlooked due to the limited data. Future studies should focus on comparing various adolescents age-groups with each other.

Furthermore, the variance in learning rates may be relevant for interpreting both social[7] and cognitive outcomes and for understanding neurodevelopmental and anxiety disorders[17]. Conversely, the rapid acquisition of the association may also indicate heightened anxiety vulnerability[20]. Disturbances in the mechanisms of associative learning during early life may impact the growth and maturity of higher-order social cognition that becomes apparent later on. The substantial number of non-learners and significant variation in learning speed warrants investigation into developmental disparities between learners and non-learners across age-groups, and the linked developmental outcomes.

## Limitations

Although eyeblink conditioning is a valuable tool for understanding basic learning mechanisms of associative learning, it is essential to consider the extent to which these findings generalise to other forms of associative learning. For example, although eyeblink conditioning and fear conditioning have some shared neural substrates, including the cerebellum, the amygdala is the primary neural circuit involved in fear conditioning. In trace eyeblink conditioning where tone and air puff are not overlapping as in delay eyeblink conditioning, the hippocampus is involved, making it difficult for infants to learn the associations due to the immaturity of the hippocampal circuitry[53,54]. It would be beneficial for future studies to investigate whether comparable age effects can be observed in trace conditioning or fear conditioning paradigms because of the different maturation of underlying neural circuitries.

Moreover, even within the context of eyeblink conditioning paradigms, there are significant variations in methodology. First, there are differences in the number of acquisition sessions. While there was often one acquisition session in adults, infants at 4 and 5 months of age required a second acquisition session to demonstrate successful conditioning[34]. Second, there are differences in the number of paired trials during acquisition. More pairings seemed to benefit infants and children in order to show successful learning and eventually reach adult levels[37]. Third, there are differences in the length of the delay between tone and air puff onset (i.e., inter-stimulus interval). For example, infants at 4–5 months of age needed a longer inter-stimulus interval (500 ms) than adults to show successful learning[36]. Thus, these factors have the potential to exert an influence on the rate and quantity

of learning in different age-groups. Accordingly, the present results can only be interpreted in the context of the parameters of the paradigm.

## Conclusions

This study demonstrates that associative learning is present across a wide range of ages, with distinct patterns in learning rate and quantity. The observed inverted U-shaped learning curve suggests that factors beyond chronological age, such as cognitive development and task demands, influence learning across the lifespan. Future research should explore the underlying mechanisms behind these findings.

Systematic study of typical age-related associative learning during development right up to adulthood is the foundation for understanding the abnormalities in associative learning, underlying the later development of a diverse range of psychopathologies. The next step is to replicate the findings presented here and to conduct fMRI studies to investigate whether there is a correlation between cerebellar brain development or connectivity and associative learning performance. As a result, exploring the development and individual differences in associative learning can potentially impact theories on aetiology and treatment options of clinical psychology and neurosciences in the future.

## Funding

## Data availability
The datasets generated during and/or analysed during the current study are available on osf: https://osf.io/nwuzx/files/osfstorage/673b5174f413dae594b28840.

## Code availability
The R-code used to analyse the data in this manuscript is available on OSF: https://osf.io/nwuzx/files/osfstorage/673b51676ded28c89ad4e1a0.

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

## Acknowledgements

Gefördert durch die Deutsche Forschungsgemeinschaft (DFG)—Projektnummer 316803389—SFB 1280. Funded by the Deutsche Forschungsgemeinschaft (DFG, German Research Foundation)—Projektnummer 316803389—SFB 1280, subproject A16, Silvia Schneider and Sarah Weigelt. The study was additionally supported by the FBZ Research Fund. :The funders had no role in study design, data collection and

analysis, decision to publish or preparation of the manuscript. Thank you to all the families who participated in the study.

## Author contributions

Carolin Konrad: Methodology, Investigation, Formal Analysis, Writing—Original Draft Preparation. Lina Neuhoff: Investigation, Writing—Review & Editing. Dirk Adolph: Conceptualisation, Methodology, Software, Writing—Review & Editing. Stephan Goerigk: Formal Analysis, Writing—Review & Editing. Jane S. Herbert: Methodology, Writing—Review & Editing. Julie Jagusch-Poirier: Investigation, Writing—Review & Editing. Sarah Weigelt: Funding Acquisition, Conceptualisation, Writing—Review & Editing. Sabine Seehagen: Methodology, Writing—Review & Editing. Silvia Schneider: Funding Acquisition, Conceptualisation, Methodology, Writing—Review & Editing.

## Competing Interests

The authors declare no competing interests.
