## [Transparent Peer Review file · Communications Psychology]

Associative Learning via Eyeblink Conditioning differs by Age from Infancy to Adulthood

Corresponding Author: Dr Carolin Konrad

Version 0:

Decision Letter:

Dear Dr Konrad,

Thank you for your patience during the peer-review process. Your manuscript titled "Lifelong learning: Associative Learning from Infancy to Adulthood" has now been seen by 3 reviewers, and I include their comments at the end of this message. They find your work of interest but raised some important points. We are interested in the possibility of publishing your study in Communications Psychology, but would like to consider your responses to these concerns and assess a revised manuscript before we make a final decision on publication.

We hope you will find the Reviewers' comments useful as you decide how to proceed. Should additional work allow you to address these criticisms, we would be happy to look at a substantially revised manuscript. If you choose to take up this option, please highlight all changes in the manuscript text file, and provide a detailed point-by-point reply to the reviewers.

Editorially, we consider it important that the revised manuscript addresses Reviewer 1's request for an additional analyses considering the effect of timing of the eyeblink responses on the findings, as well as Reviewer 3's request for additional analyses examining the sensitivity of the results to the removal of non-learners. Both Reviewer 2 and Reviewer 3 request a better literature review regarding associative learning and how your study is situated in the literature.

Please make sure the revised manuscript also includes information on participant consent/assent.

I am attaching a checklist that details critical reporting requirements for the revised manuscript. Please attend to each item and ensure your manuscript is fully compliant. We are requesting that your manuscript aligns with these requirements as this facilitates the evaluation of your manuscript, reducing delays in re-review and potential future acceptance. If your revised manuscript is not aligned with these requests on major issues, such as those concerning statistics, it may be returned to you for further revisions without re-review. Additional information can be found in our style and formatting guide https://www.nature.com/documents/commspsychol-style-formatting-guide-accept.pdf Communications Psychology formatting guide.

If the revision process takes significantly longer than five months, we will be happy to reconsider your paper at a later date, provided it still presents a significant contribution to the literature at that stage.

Please use the following link to submit your
- revised manuscript,
- point-by-point response to the referees' comments,

- cover letter (as a separate document),
- the Editorial Policy Checklist (see below),
- the Reporting Summary (see below), and
- the completed Editorial Request Table (attached):

Link Redacted

Thank you for the opportunity to review your work.

Best regards,

Jennifer Bellingtier

Jennifer Bellingtier, PhD
Senior Editor
Communications Psychology

REVIEWER EXPERTISE:

Reviewer #1 learning, conditioning
Reviewer #2 development, learning
Reviewer #3 conditioning, development

REVIEWER REPORTS:

Reviewer #1 (Remarks to the Author):

In this paper, the authors present the outcomes of a study involving several hundred participants ranging in age from 12 months to 29 years. All participants underwent a two-day eyeblink conditioning protocol.

Over the course of two days, the participants were subjected to a total of 48 paired trials. The interstimulus interval, which the authors should state more explicitly was set at 300 milliseconds. The overall learning rate was somewhat disappointing, barely surpassing 53% on the second day.

The study yielded several intriguing and significant findings. Firstly, the results indicate that school-aged children outperform both infants and adults in learning, a finding that contradicts previous studies. Another noteworthy observation is the substantial variance in the learning rate across all age groups, with a considerable number of participants showing minimal learning, while others demonstrated rapid learning.

The authors deserve praise for their innovative and informative illustrations, which efficiently present a wealth of data, thereby highlighting the highly variable learning rate (see Figure 1 and Figure 3).

The study's unusually large sample size, combined with these compelling observations, makes this paper a valuable addition to the existing literature. Therefore, I recommend its publication. However, certain aspects require further clarification and/or justification.

My primary concern is the authors' use of an unconventional training protocol. The training protocol could account for the discrepancies observed between their results and those of other studies. For example, the authors opted for a lower number of trials (48 over two days), whereas most human studies typically employ 60-100 trials in a single day. The authors need to justify their deviation from the standard protocol and discuss its potential impact on the results.

The choice of a 300ms interval also warrants further explanation. Most human studies use a longer interstimulus interval, and evidence suggests that longer intervals lead to improved learning (source). Could the relatively low learning rate be attributed to the short ISI?

It is also peculiar that 12-month-olds outperform older infants, yet primary school children perform best overall. This suggests that learning ability initially declines after 12 months, then increases upon reaching primary school age before declining again. The authors speculate that primary school children excel due to their extensive learning requirements. However, why do 12-month-olds outperform 24-month-olds?

One aspect of eyeblink conditioning that is conspicuously absent is the timing of the responses, which could be a crucial variable. This is particularly relevant since the authors have classified all eyeblinks within the interstimulus interval as conditioned responses. This classification could lead to the misidentification of startle responses as conditioned responses. It is also plausible that infants exhibit more startle responses or a higher baseline blink rate than older infants and adults.

Typically, researchers do not count blink responses occurring within the first 100ms of the ISI, as such responses are likely to be startle or spontaneous blinks.

These issues are important and should be addressed through further data analysis or in the paper's discussion. Despite these concerns, the results contribute a valuable piece to the puzzle and are worthy of publication.

Reviewer #2 (Remarks to the Author):

Summary: A conditioned eyeblink procedure that was completed across 2 sessions occurring on consecutive days was used in infants, primary school-aged children, adolescents and young adults to explore possible age-related changes in associative learning. Learning was observed in all age groups, but primary school-aged children displayed the most consistent and least variable learning. Adults and adolescents exhibited faster association learning than infants. An additional learning session (on day 2) supported learning in infants and adolescents (but not children or adults). Interesting differences in the patterns of learning were also discovered using a clustering analysis: infants demonstrated either rapid learning, with significant improvements from day 1 to day 2, or no learning; primary school-aged children exhibited either immediate learning or a rapid increase in learning; adolescents showed the most diverse distribution of learning types; and adults, demonstrated either a lack of learning or fast-paced learning.

Evaluation: I think this paper is interesting and important and could have a very wide appeal! It's very interesting to see an example of better learning in childhood as compared to adulthood; and the age coverage – from infancy to adulthood – is commendable! I also think the paradigm is nicely executed, and the analyses are done comprehensively and well. I offer the following suggestions in the hope of improving the paper.

Suggestions/Questions

1. Most pressing, I think the framing of this paper needs to be revised. I know that there is not a lot of space in this journal and that the intro and discussion are quite short, but I think it's critical to indicate specifically what you are referring to by associative learning in order to identify the gaping hole. There are other experimental paradigms that are not conditioning (or eye blink conditioning specifically), but that measure associative learning. You cite the Amso & Davidow (2012) paper which looks at a form of associative learning (cue-target pairings that are probabilistic) in a wide age range. I also think associative-memory paradigms and even statistical learning and SRT paradigms (e.g., Janacsek, Fiser, & Nemeth, 2012) are relevant here. It's very important to note how this work extends/and is different from this and I think clearly defining the kind of learning you are looking at and how it is unique will be really important to do in the introduction. There is a clear argument to be made for needing the conditioned eye blink data to enhance this picture, it just isn't yet clearly spelled out. To make space for this, I might suggest removing the multiple references to atypical development since this isn't studied in the current manuscript.

2. I was also a little confused about the chosen sample sizes and the stopping rules. It sounds like this study was pre-registered, but I did not see a link to this for my review, so I could not check myself. First, what did you pre-register? All of the reported analyses and sample sizes and exclusion criteria? On page 17, you note a stopping rule, but the data collection spans a super long time. Was that date selected in advance?

Reviewer #3 (Remarks to the Author):

The authors of Lifelong learning: Associative learning from infancy through adulthood present results of a cross-sectional study in which they measured associative learning of a tone and air-puff by measuring conditioned eye-blink responses over two days in infants, children, adolescents, and young adults (although not older adults as maybe implied by the title). They examined learning rate and different learning trajectories in each group, and directly compared the age groups. They have a number of conclusions, including that children (7-9-years-old) learned fastest, that adolescents were the most variable in their learning trajectories, and that infants relied on consolidation mechanisms to demonstrate associative learning while other age groups did not. The data presented herein have the potential to be an important contribution but without a great deal more background on the current state of the field, and more explanation of the results, I worry they are too vague for the readership of Communications Psychology. I elaborate on these points below, and provide some suggestions the authors may wish to consider.

1. The authors state multiple times that this is the first investigation into associative learning across the lifespan, but cite work examining associative learning across different age ranges. What would the extant literature on the development of associative learning with different paradigms lead us to believe would be the developmental trajectory within one paradigm? I appreciate that completing a developmental study with one paradigm is useful insight, but there surely must be background literatures that are relevant to discuss. In particular, it seems like the authors have chosen to restrict their discussion of prior work to that which used the same paradigm they have chosen, rather than work about the cognitive construct in general. Could they provide a clear definition of associative learning that makes it obvious why some previous literatures are or are not relevant? At the moment that is not clear. There is also a great deal of work on the development of associative memory (or a number of other cognitive processes) in infancy, childhood, and adolescence which might be relevant to discuss here, and which could help situate the current work in a broader developmental literature. Relatedly, why do the authors think

these differences are emerging? What might be the underlying reason for these changes? Much more interpretation and explanation of your conclusions is needed for your readers to understand why you are suggesting the claims you are making.

2. It is also not clear how these results immediately link to many of the clinical applications the authors bring up. Can they please elaborate significantly (or replace these speculations with interpretations more closely related to the data?)

3. It is tricky to understand some of the results, especially prior to reading the methods. Even in the methods, it is not obvious how many of the analyses are in the supplementary material versus explained in the paper. For example, in the results section it is not clear what is being modelled differently in the learning speed and learning quantity sections, or why these metrics might show different results theoretically. Later in the methods these analyses are more specified, but upon reading the results much is left unclear. Some explanation of why the authors are reporting each result and what mathematical approach was involved in each section be very helpful for understanding these results as readers are making their way through the paper. Similarly, some additional clarity on the patterns of results across age groups, rather than just pairwise comparison reporting would go a long way in helping readers understand what the take home message from all of these results are.

Some smaller questions:

4. Do the authors think that the heterogeneity in the adolescent group's learning clusters is because of the very wide age range included here? If they examine learning cluster as a function of age are the younger adolescents more likely to show child-like trajectories, for example? In regard to the sample, I wonder if the authors could acknowledge or explain any potential reasons that the sparse sampling across childhood, and especially adolescence might have contributed to their results (i.e. they have ~25-30 people every 12 months in the first three years of life, but 30 people total from age 12-17.

5. Relatedly, if you remove non-learners (who failed to acquire the association) from the analysis, do your results hold? What does that suggest about the development of associative learning?

6. As a stylistic note, many of the grey-scale figures (e.g. Figure 1a, Fig 3) are quite hard to read because it is very hard to make out the difference between grey-scale values (i.e. missing data vs. no blink)

Typos:

Line 208, cluster should read clusters

Line 436, test should read tests

Line 438, gliding average... should this be 'sliding' average?

EDITORIAL POLICIES

EDITORIAL POLICIES

We ask that you ensure your manuscript complies with our editorial policies and reporting requirements.

To that end, we require revised manuscripts to be accompanied by two completed items: a reporting summary that collects information on study design and procedure, and an editorial policy checklist that verifies compliance with all required editorial policies

- <https://www.nature.com/documents/nr-reporting-summary.zip> Nature Research Reporting Summary
- <https://www.nature.com/documents/nr-editorial-policy-checklist.pdf> Editorial Policy Checklist

All points on the policy checklist must be addressed. Your revised manuscript can only be sent back to the referees if these checklists are completed and uploaded with the revision.

Notes: If you have submitted a Stage 1 Registered Report, Review, Primer, Comment, or Perspective you do not need to submit these forms. If you have already submitted these forms, you may disregard this request.

* TRANSPARENT PEER REVIEW: Communications Psychology uses a transparent peer review system. This means that we publish the editorial decision letters including Reviewers' comments to the authors and the author rebuttal letters online

as a supplementary peer review file. However, on author request, confidential information and data can be removed from the published reviewer reports and rebuttal letters prior to publication. If your manuscript has been previously reviewed at another journal, those Reviewers' comments would not form part of the published peer review file.

Communications Psychology is committed to improving transparency in authorship. As part of our efforts in this direction, we are now requesting that all authors identified as 'corresponding author' create and link their Open Researcher and Contributor Identifier (ORCID) with their account on the Manuscript Tracking System prior to acceptance. ORCID helps the scientific community achieve unambiguous attribution of all scholarly contributions. You can create and link your ORCID from the home page of the Manuscript Tracking System by clicking on 'Modify my Springer Nature account' and following the instructions in the link below. Please also inform all co-authors that they can add their ORCID to their accounts and that they must do so prior to acceptance.

If you experience problems in linking your ORCID, please contact the Platform Support Helpdesk.

Version 1:

Decision Letter:

Dear Dr Konrad,

Your manuscript titled "Lifelong learning: Associative Learning from Infancy to Adulthood" has now been seen by our reviewers, whose comments appear below. In light of their advice I am delighted to say that we are happy, in principle, to publish a suitably revised version in Communications Psychology.

We therefore invite you to revise your paper one last time to address the remaining concerns of our reviewers and a list of editorial requests. At the same time we ask that you edit your manuscript to comply with our format requirements and to maximise the accessibility and therefore the impact of your work.

EDITORIAL REQUESTS:

SUBMISSION INFORMATION:

OPEN ACCESS:

* CODE AVAILABILITY: All Communications Psychology manuscripts must include a section titled "Code Availability" at the end of the methods section. We require that the custom analysis code supporting your conclusions is made available in a publicly accessible repository at this stage; please choose a repository that generates a digital object identifier (DOI) for the code; the link to the repository and the DOI must be included in the Code Availability statement. Publication as Supplementary Information will not suffice.

* DATA AVAILABILITY:

Link Redacted

Best regards,

Jennifer Bellingtier

Jennifer Bellingtier, PhD
Senior Editor
Communications Psychology

REVIEWERS' EXPERTISE:

Reviewer #1 learning, conditioning
Reviewer #3 conditioning, development

REVIEWERS' COMMENTS:

Reviewer #1 (Remarks to the Author):

I appreciate the authors' significant efforts in revising the manuscript based on the previous reviews. Most of the concerns I initially raised have been satisfactorily addressed, and key misunderstandings—such as those related to the ISI—have been clarified. As noted in my earlier review, this dataset represents a valuable contribution to the field, offering insights that will undoubtedly interest researchers. However, despite these substantial improvements, a few issues still need attention before I can recommend the manuscript for publication.

Major Concerns

1. Training Paradigm:

In my previous review, I emphasized that the differences between the authors' training paradigm and the more commonly used paradigms were not sufficiently discussed. Given that variations in training paradigms can significantly affect the outcomes, this should be highlighted. I recommend that the authors briefly describe the paradigm (e.g., 48 trials over two days) in the abstract and discuss its potential implications more explicitly in the discussion section.

2. Associative Learning:

In addition, I noticed that my earlier comments may not have fully conveyed the need for specificity in describing the experimental paradigm in the abstract. Eyeblink conditioning is framed as a general measure of associative learning, which seems too broad. Associative learning encompasses various forms that depend on different brain regions, and the authors should clarify how their findings relate to this broader concept. I suggest that the discussion touches on whether and to what extent their findings might generalize to other forms of associative learning.

Minor Details

1. Clarification of Timing:

The authors should specify what is meant by the “timing” of the conditioned responses. For instance, are they referring to the onset of the blink response, the peak, or some other measure? For example: “Mean latency of the conditioned response on day 1 was 507 ms (SD = 28 ms) for 12-month-olds, 511 ms (SD = 87 ms) for 18-month-olds, and 533 ms (SD = 79 ms)...”

2. Neutral Language in Results:

In the results section, the statement that “blink responses with shorter latencies are more well-timed” could be misleading. Many researchers might argue that a blink response that occurs just before the airpuff is more accurately “timed.” I recommend using more neutral language to avoid this interpretation. It would suffice to state that blink responses occurred earlier on the second day. For instance: “A post-hoc paired t-test revealed that blink latencies were significantly shorter on day 2 than on day 1, $t(172) = 3.55$, $p < .001$, $d = .270$.”

Reviewer #3 (Remarks to the Author):

Thanks to the authors for their revision—I was happy to see the figures are more legible, and that the introduction is a bit more comprehensive. I also think that moving the methods section to be ahead of the results was helpful. This makes the main take-home messages, that children show uniquely fast learning and infants alone needed a consolidation period to show learning, more obvious. These findings are interesting and will be of interest to developmental psychologists. That said, I still found the results section a bit hard to follow. Ultimately the clarity of the results section, and how they chose to organize their findings, is up to the authors. I will include below some findings which I think could be presented more clearly if the authors revise the manuscript again.

- 1) The authors present the %CR response and learning rate analyses in an interleaved way which is not intuitive. This is because they have a stand-alone ‘preregistered analysis’ section, but this has had the effect of confusing the reader about which metric is being discussed when, and how they differ.
- 2) Relatedly, I asked in my first round of comments about the difference between learning rate and performance-- I understand why someone would want to look at learning rate separately from performance. But the way these separate questions are described is still pretty overlapping. For instance, after the pre-registered analysis section, there is an analysis which includes ‘block’ in an ANOVA analyzing increases in the CR. This therefore describes when during learning differences are emerging. Then, there is a section about trial-by-trial analyses of learning rate, where it is not clear what the difference in learning speed is reporting (Is this a subject-specific slope of the effect of trial number?) or how this is answering a theoretically separate question from the prior analyses.
- 3) In the last round of comments, I asked the authors whether their results hold if they exclude non-learners. In the paragraph the authors added to address this, the answer seems to be that the conclusions do change—for example, adolescents now also benefitted from a second session, like the infants originally? I am not sure why infants were collapsed into one group for this section alone, and I am again not sure what the depended variable for ‘learning rate’ here is. It also This is also the first place where improvement is described as a percentage. Can the authors more clearly state whether their main results change if non-learners are excluded?
- 4) The authors also did not address my question of whether the variable age range in the adolescent group accounted for their effects. They added a note to the discussion that says someone else could look at this, but this is also an analysis they could run directly.

Responses to the reviewers

Reviewer #1:

In this paper, the authors present the outcomes of a study involving several hundred participants ranging in age from 12 months to 29 years. All participants underwent a two-day eyeblink conditioning protocol. Over the course of two days, the participants were subjected to a total of 48 paired trials. The interstimulus interval, which the authors should state more explicitly was set at 300 milliseconds. The overall learning rate was somewhat disappointing, barely surpassing 53% on the second day.

The study yielded several intriguing and significant findings. Firstly, the results indicate that school-aged children outperform both infants and adults in learning, a finding that contradicts previous studies. Another noteworthy observation is the substantial variance in the learning rate across all age groups, with a considerable number of participants showing minimal learning, while others demonstrated rapid learning.

The authors deserve praise for their innovative and informative illustrations, which efficiently present a wealth of data, thereby highlighting the highly variable learning rate (see Figure 1 and Figure 3). The study's unusually large sample size, combined with these compelling observations, makes this paper a valuable addition to the existing literature. Therefore, I recommend its publication. However, certain aspects require further clarification and/or justification.

1. My primary concern is the authors' use of an unconventional training protocol. The training protocol could account for the discrepancies observed between their results and those of other studies. For example, the authors opted for a lower number of trials (48 over two days), whereas most human studies typically employ 60-100 trials in a single day. The authors need to justify their deviation from the standard protocol and discuss its potential impact on the results.

Response: Our objective was to identify a paradigm that could be applicable to individuals across a wide age range. We chose a training protocol that also has been tested in infant populations. Previous studies have shown that infants require a second acquisition session on a different day. Therefore, we employed the same design used in other eyeblink conditioning studies with infants (e.g., Herbert et al., 2003; Ivkovich et al., 1999) to ensure that even the youngest participants in our study are able to demonstrate successful learning. As eyeblink conditioning can be challenging in infants due to factors such as the air puff and their inability to remain seated for long periods, it is not feasible to use the same number of trials in a single day as in adults. Therefore, the results can only be directly compared with other studies involving infants and children that also included adults as a control condition.

2. The choice of a 300ms interval also warrants further explanation. Most human studies use a longer interstimulus interval, and evidence suggests that longer intervals lead to improved learning (source). Could the relatively low learning rate be attributed to the short ISI?

Response: We apologize for any confusion regarding the inter-stimulus interval and thank the reviewer for bringing this to our attention. We used a 750 ms tone and a 100 ms airpuff, and, as in other infant and adult studies, we employed a 650 ms inter-stimulus interval to ensure successful learning. We have now clearly stated the inter-stimulus interval in the method section.

3. It is also peculiar that 12-month-olds outperform older infants, yet primary school children perform best overall. This suggests that learning ability initially declines after 12 months, then

increases upon reaching primary school age before declining again. The authors speculate that primary school children excel due to their extensive learning requirements. However, why do 12-month-olds outperform 24-month-olds?

Response: The reviewer raises an excellent point regarding the observed peak in performance at 12 months. We acknowledge that this finding warrants further discussion. We analysed how often infants reacted to the air puff in the air puff alone trials and whether this was related to how quickly they learned the association between the tone and the air puff. We found a negative correlation only in 12-month-olds, indicating that the more they reacted to the air puff, the fewer trials they needed to show successful learning. This suggests that 12-month-olds may have perceived the air puff as more aversive than other age groups of infants. However, when only learners are considered, 12-month-olds exhibit superior performance on day 1 compared to 24-month-olds, yet demonstrate comparable learning rates to 18- and 36-month-olds. On day 2, 24-month-olds demonstrate a greater increase in conditional responses than 12-month-olds. Therefore, the performance of 12-month-olds can be attributed to the low rate of non-learners and the high rate of non-learners in other infant age groups. The reason for the significant discrepancy in the number of learners and non-learners remains a topic for further investigation. We revised the manuscript to address this observation by acknowledging the temporary dip in performance at 24 months in the discussion section.

4. One aspect of eyeblink conditioning that is conspicuously absent is the timing of the responses, which could be a crucial variable. This is particularly relevant since the authors have classified all eyeblinks within the interstimulus interval as conditioned responses. This classification could lead to the misidentification of startle responses as conditioned responses. It is also plausible that infants exhibit more startle responses or a higher baseline blink rate than older infants and adults. Typically, researchers do not count blink responses occurring within the first 100ms of the ISI, as such responses are likely to be startle or spontaneous blinks.

Response: We have revised the method section to clarify that the interval defined as a conditioned response excluded the startle period. Specifically, we defined a conditioned response as an eyeblink occurring between 300ms after tone onset until the air puff onset, to avoid startle reactions. While past research in studies with human adults used a duration of the startle period of 100-200 ms, the increased duration in our developmental studies reflects the potentially slower response latencies of infants, in accordance with previous eyeblink conditioning studies in infants (e.g., Herbert et al., 2003; Ivkovich et al., 1999). Furthermore, we have now also analyzed the timing of all conditioned responses and added these findings to the results section. There were differences in the timing of the conditioned responses between age-groups, but conditioned responses were overall better timed on day 2 than on day 1.

Reviewer #2:

Summary: A conditioned eyeblink procedure that was completed across 2 sessions occurring on consecutive days was used in infants, primary school-aged children, adolescents and young adults to explore possible age-related changes in associative learning. Learning was observed in all age groups, but primary school-aged children displayed the most consistent and least variable learning. Adults and adolescents exhibited faster association learning than infants. And an additional learning session (on day 2) supported learning in infants and adolescents (but not children or adults). Interesting differences in the patterns of learning were also discovered using a clustering analysis: infants demonstrated either rapid learning, with significant improvements from day 1 to day 2, or no

learning; primary school-aged children exhibited either immediate learning or a rapid increase in learning; adolescents showed the most diverse distribution of learning types; and adults, demonstrated either a lack of learning or fast-paced learning.

Evaluation: I think this paper is interesting and important and could have a very wide appeal! It's very interesting to see an example of better learning in childhood as compared to adulthood; and the age coverage – from infancy to adulthood – is commendable! I also think the paradigm is nicely executed, and the analyses are done comprehensively and well. I offer the following suggestions in the hope of improving the paper.

Suggestions/Questions

1. Most pressing, I think the framing of this paper needs to be revised. I know that there is not a lot of space in this journal and that the intro and discussion are quite short, but I think it's critical to indicate specifically what you are referring to by associative learning in order to identify the gaping hole. There are other experimental paradigms that are not conditioning (or eye blink conditioning specifically), but that measure associative learning. You cite the Amso & Davidow (2012) paper which looks at a form of associative learning (cue-target pairings that are probabilistic) in a wide age range. I also think associative-memory paradigms and even statistical learning and SRT paradigms (e.g., Janacek, Fiser, & Nemeth, 2012) are relevant here. It's very important to note how this work extends/and is different from this and I think clearly defining the kind of learning you are looking at and how it is unique will be really important to do in the introduction. There is a clear argument to be made for needing the conditioned eye blink data to enhance this picture, it just isn't yet clearly spelled out. To make space for this, I might suggest removing the multiple references to atypical development since this isn't studied in the current manuscript.

Response: A clear definition of associative learning is now provided in the introduction. It should be noted that associative learning is a broad category that encompasses many of our daily learning activities that involve forming associations between stimuli and/or responses. It is commonly divided into classical conditioning and operant conditioning (e.g., Byrne et al., 2014). Therefore, we now describe studies on the development of operant conditioning as well. We then propose the additional value of using eyeblink conditioning in our study and also added developmental studies from different age-groups in the introduction.

2. I was also a little confused about the chosen sample sizes and the stopping rules. It sounds like this study was pre-registered, but I did not see a link to this for my review, so I could not check myself. First, what did you pre-register? All of the reported analyses and sample sizes and exclusion criteria? On page 17, you note a stopping rule, but the data collection spans a super long time. Was that date selected in advance?

Response: The study was funded for a period of four years. Therefore, resources for testing finished in May 2021 and the date (stopping rule) was selected in advance. The preregistered study can be found at https://osf.io/cvnge/?view_only=008c21c243c843f7a9c4cef0b0a96350

We preregistered the exclusion criteria, planned sample size, and analyses before conducting any data analyses. However, we decided to conduct a more detailed data analysis on a trial-by-trial basis afterwards. Therefore, we included both the preregistered analyses on blocks of trials using common methods, as in other papers, and the more detailed and innovative approach in the results section. We have now marked the preregistered analyses and the additional analyses more clearly in the manuscript. The additional analyses do not alter the overall pattern of results or message of the

paper, but rather paint a more detailed picture of the data.

Reviewer #3:

The authors of Lifelong learning: Associative learning from infancy through adulthood present results of a cross-sectional study in which they measured associative learning of a tone and air-puff by measuring conditioned eye-blink responses over two days in infants, children, adolescents, and young adults (although not older adults as maybe implied by the title). They examined learning rate and different learning trajectories in each group, and directly compared the age groups. They have a number of conclusions, including that children (7-9-years-old) learned fastest, that adolescents were the most variable in their learning trajectories, and that infants relied on consolidation mechanisms to demonstrate associative learning while other age groups did not. The data presented herein have the potential to be an important contribution but without a great deal more background on the current state of the field, and more explanation of the results, I worry they are too vague for the readership of Communications Psychology. I elaborate on these points below, and provide some suggestions the authors may wish to consider.

1. The authors state multiple times that this is the first investigation into associative learning across the lifespan, but cite work examining associative learning across different age ranges. What would the extant literature on the development of associative learning with different paradigms lead us to believe would be the developmental trajectory within one paradigm? I appreciate that completing a developmental study with one paradigm is useful insight, but there surely must be background literatures that are relevant to discuss. In particular, it seems like the authors have chosen to restrict their discussion of prior work to that which used the same paradigm they have chosen, rather than work about the cognitive construct in general. Could they provide a clear definition of associative learning that makes it obvious why some previous literatures are or are not relevant? At the moment that is not clear. There is also a great deal of work on the development of associative memory (or a number of other cognitive processes) in infancy, childhood, and adolescence which might be relevant to discuss here, and which could help situate the current work in a broader developmental literature.

Relatedly, why do the authors think these differences are emerging? What might be the underlying reason for these changes? Much more interpretation and explanation of your conclusions is needed for your readers to understand why you are suggesting the claims you are making.

Response: To ensure clarity, we provide a more precise definition of associative learning: Associative learning refers to the process of forming connections between stimuli or events, such that the presence of one element triggers the expectation or retrieval of the other. Two principal forms of associative learning exist: classical conditioning and operant conditioning (Byrne et al., 2014). We added more studies on the research of associative memory in infancy, childhood, and adolescence for both classical and operant conditioning. In our study, we chose to focus on the eyeblink conditioning because it offers some unique advantages. It does not require a complex behavioural or verbal response, making it suitable for studying infants. Furthermore, eyeblink conditioning has a highly conserved neural circuitry and is dependent on the cerebellum, offering a window into early brain development.

However, we recognize the value of understanding the broader developmental picture. Therefore, we have included into our discussion of the literature to include relevant findings from studies employing different implicit learning paradigms. The developmental trajectory of implicit learning remains debated – with some studies supporting an "invariant hypothesis" of no change or

improvement across development (e.g., Amso & Davidow, 2012; Kartekin et al., 2006; Muelemans et al., 1998) and others suggesting age-related improvements (e.g., Janacsek et al., 2012; Vaiydia et al., 2007). We have expanded the interpretation of our results to explore potential explanations for the observed age-related differences. These explanations might involve factors underlying the observed age differences such as changes in cognitive processing speed, motivational differences, or sensitivity to the air puff. We additionally deleted interpretations and clinical applications not warranted by the data, as requested by the editor.

2. It is also not clear how these results immediately link to many of the clinical applications the authors bring up. Can they please elaborate significantly (or replace these speculations with interpretations more closely related to the data?)

Response: We appreciate the reviewer's comment regarding the connection between our results and the discussed clinical applications. We agree that further research is needed to explore the connection between our findings and the observed clinical phenomenon. We have adjusted the discussion section to focus on the well-supported applications and removed speculative statements that were not directly linked to the data.

3. It is tricky to understand some of the results, especially prior to reading the methods. Even in the methods, it is not obvious how many of the analyses are in the supplementary material versus explained in the paper. For example, in the results section it is not clear what is being modelled differently in the learning speed and learning quantity sections, or why these metrics might show different results theoretically. Later in the methods these analyses are more specified, but upon reading the results much is left unclear. Some explanation of why the authors are reporting each result and what mathematical approach was involved in each section be very helpful for understanding these results as readers are making their way through the paper. Similarly, some additional clarity on the patterns of results across age groups, rather than just pairwise comparison reporting would go a long way in helping readers understand what the take home message from all of these results are.

Response: The paper's format has been changed to a classical style, with the method section preceding the results section. We have also included additional sentences to explain the reasoning behind the analyses conducted and highlighted the specific analyses on which the results are based.

It is now clearly marked in the method section what is being modelled in the learning speed and quantity analyses. The learning speed can be defined as the rate at which the association between conditioned and unconditioned stimuli strengthens. The quantity of learning is defined as the overall level of association achieved following training. It is necessary distinguish between the two metrics. For instance, participant may exhibit rapid initial learning (high learning speed) but may eventually reach a lower level of association (lower learning quantity). Conversely, other participants may demonstrate a slower initial learning rate but ultimately achieve a higher overall level of association. Therefore, we included analyses on both the learning speed and quantity in the results section.

Furthermore, we have added information on the pattern of results across age-groups to the discussion section to emphasize the key message for each group, as suggested by the reviewer.

4. Do the authors think that the heterogeneity in the adolescent group's learning clusters is because of the very wide age range included here? If they examine learning cluster as a function of age are the younger adolescents more likely to show child-like trajectories, for example? In regard to the sample, I wonder if the authors could acknowledge or explain any potential reasons that the sparse sampling across childhood, and especially adolescence might have contributed to their

results (i.e. they have ~25-30 people every 12 months in the first three years of life, but 30 people total from age 12-17.

Response: Given the absence of data on children aged 1-3 years and the rapid pace of development during this period, we were particularly interested in age-related changes in infancy. To this end, we opted for a relatively small window of testing infants and multiple samples. Given the existing body of research on adolescents, we chose to have one sample during this period. However, we recognize that this may limit our ability to gain a more detailed understanding of age-related changes during adolescence. The limited sampling in adolescence could have a significant impact on the results. There is a possibility that subgroups within the adolescent age range may be overlooked due to the limited data. We acknowledge this limitation and discuss the potential consequences. It is recommended that future research be conducted with a more robust sample size in the adolescent age group.

5. Relatedly, if you remove non-learners (who failed to acquire the association) from the analysis, do your results hold? What does that suggest about the development of associative learning?

Response: We have conducted additional analyses on learning rates for learners only. The results obtained from the 7- to 8-year-old children, adolescents, and adults were consistent. However, the finding that infants in the learner group have steeper learning curves than adults is particularly intriguing. A possible interpretation of this finding is that infants who successfully acquired the association may have exhibited a stronger initial response to the airpuff, which could have resulted in a steeper initial learning curve. This could be attributed to heightened sensitivity to novelty or a greater focus on establishing new associations during this critical developmental period.

6. As a stylistic note, many of the grey-scale figures (e.g. Figure 1a, Fig 3) are quite hard to read because it is very hard to make out the difference between grey-scale values (i.e. missing data vs. no blink)

Response: We thank the reviewer for this comment and we changed the colors of the figures.

7. Typos:

Line 208, cluster should read clusters

Line 436, test should read tests

Line 438, gliding average... should this be 'sliding' average?

Response: We thank the reviewer for pointing out these typos and corrected them. We changed the term "gliding average" to "moving average".

Reviewer #1:

I appreciate the authors' significant efforts in revising the manuscript based on the previous reviews. Most of the concerns I initially raised have been satisfactorily addressed, and key misunderstandings—such as those related to the ISI—have been clarified. As noted in my earlier review, this dataset represents a valuable contribution to the field, offering insights that will undoubtedly interest researchers. However, despite these substantial improvements, a few issues still need attention before I can recommend the manuscript for publication.

Major Concerns

1. Training Paradigm:

In my previous review, I emphasized that the differences between the authors' training paradigm and the more commonly used paradigms were not sufficiently discussed. Given that variations in training paradigms can significantly affect the outcomes, this should be highlighted. I recommend that the authors briefly describe the paradigm (e.g., 48 trials over two days) in the abstract and discuss its potential implications more explicitly in the discussion section.

Response: To address these concerns, we have made the following revisions:

Abstract: We have added a sentence to the abstract describing the paradigm: "We employed a classical delay eyeblink conditioning paradigm that consisted of two sessions with a total of 48 paired trials "

Discussion: We have expanded the discussion section to include an analysis of the potential implications of our paradigm. We have compared our approach with other paradigms examining associative learning.

2. Associative Learning:

In addition, I noticed that my earlier comments may not have fully conveyed the need for specificity in describing the experimental paradigm in the abstract. Eyeblink conditioning is framed as a general measure of associative learning, which seems too broad. Associative learning encompasses various forms that depend on different brain regions, and the authors should clarify how their findings relate to this broader concept. I suggest that the discussion touches on whether and to what extent their findings might generalize to other forms of associative learning.

Response: As mentioned above, we have added a sentence to the abstract describing the paradigm: "We employed a classical delay eyeblink conditioning paradigm that consisted of two sessions with a total of 48 paired trials ". We have expanded the discussion section to explore the relationship between our findings and the broader concept of associative learning. We have compared and contrasted our results with studies on other forms of associative learning, such as fear conditioning and instrumental conditioning. We have also considered the potential limitations of generalizing our findings to other forms of associative learning, acknowledging that different brain regions and behavioral responses may be involved in these processes.

Minor Details

1. Clarification of Timing:

The authors should specify what is meant by the "timing" of the conditioned responses. For instance, are they referring to the onset of the blink response, the peak, or some other measure? For example:

“Mean latency of the conditioned response on day 1 was 507 ms (SD = 28 ms) for 12-month-olds, 511 ms (SD = 87 ms) for 18-month-olds, and 533 ms (SD = 79 ms)...”

Response: We thank the reviewer for this comment and have amended the manuscript to provide a more precise definition by replacing “timing” with “onset time of conditioned responses (onset of eye lid closure)”.

2. Neutral Language in Results:

In the results section, the statement that “blink responses with shorter latencies are more well-timed” could be misleading. Many researchers might argue that a blink response that occurs just before the airpuff is more accurately “timed.” I recommend using more neutral language to avoid this interpretation. It would suffice to state that blink responses occurred earlier on the second day. For instance: “A post-hoc paired t-test revealed that blink latencies were significantly shorter on day 2 than on day 1, $t(172) = 3.55$, $p < .001$, $d = .270$.”

Response: We thank the reviewer for this comment. We changed the sentences to a neutral language in the result section: “A post-hoc paired t-test showed that the conditioned responses occurred earlier (i.e., shorter latencies) on day 2 than on day 1, $t(172) = 3.55$, $p < .001$, $d = .270$.”

Reviewer #3:

Thanks to the authors for their revision—I was happy to see the figures are more legible, and that the introduction is a bit more comprehensive. I also think that moving the methods section to be ahead of the results was helpful. This makes the main take-home messages, that children show uniquely fast learning and infants alone needed a consolidation period to show learning, more obvious. These findings are interesting and will be of interest to developmental psychologists. That said, I still found the results section a bit hard to follow. Ultimately the clarity of the results section, and how they chose to organize their findings, is up to the authors. I will include below some findings which I think could be presented more clearly if the authors revise the manuscript again.

1. The authors present the %CR response and learning rate analyses in an interleaved way which is not intuitive. This is because they have a stand-alone ‘preregistered analysis’ section, but this has had the effect of confusing the reader about which metric is being discussed when, and how they differ.

Response: We understand that this can be confusing. We would prefer to limit the scope of our analysis to the learning rate on a trial-by-trial basis, as this provides a more detailed and nuanced picture than the pre-registered analyses. Nevertheless, the journal requires that the analyses be included as preregistered as well. This is also consistent with the principles of open science. To resolve the issue, we incorporated explanatory sentences into the analysis section to differentiate between the various measures.

2. Relatedly, I asked in my first round of comments about the difference between learning rate and performance- I understand why someone would want to look at learning rate separately from performance. But the way these separate questions are described is still pretty overlapping. For instance, after the pre-registered analysis section, there is an analysis which includes ‘block’ in an ANOVA analyzing increases in the CR. This therefore describes when during learning differences are emerging. Then, there is a section about trial-by-trial analyses of learning rate, where it is not clear what the difference in learning speed is reporting (Is this a subject-specific

slope of the effect of trial number?) or how this is answering a theoretically separate question from the prior analyses.

Response: We understand your concern about the potential overlap between the analyses of learning rate and performance. We aim to clarify this distinction in the revised methods section by clarifying the analysis in each section of the data analysis and revised the headings for the result sections.

“Preregistered analyses

For the preregistered analyses, we were interested in how the amount of conditioned responses per block changed across session 1 and 2.”

“Analyses of the rate of learning

We sought to analyse the changes in the rate of learning using piecewise linear mixed modelling (LMM). This enables the evaluation of changes in conditioned responses over the course of the sessions.”

“Analyses of the quantity of learning per day

Next, the total numbers of conditioned responses per day (i.e., the amount of learning) and the timing of the conditioned responses were compared”

3. In the last round of comments, I asked the authors whether their results hold if they exclude non-learners. In the paragraph the authors added to address this, the answer seems to be that the conclusions do change—for example, adolescents now also benefitted from a second session, like the infants originally? I am not sure why infants were collapsed into one group for this section alone, and I am again not sure what the depended variable for ‘learning rate’ here is. This is also the first place where improvement is described as a percentage. Can the authors more clearly state whether their main results change if non-learners are excluded?

Response: We thank the reviewer for this comment and hope that we can clarify this point. Infants were not collapsed into one group and results for each group are visible in Tables 8 and 9. Analyses are equivalent to the ones in the subsection ***Age differences in the rate of learning***. We stated in the discussion how the pattern of result changes when only analyzing learners:

“Further analyses were conducted on the learning rates of the learners only. The results obtained from the primary school children, adolescents, and adults were found to be consistent. However, the finding that infants in the learner group exhibit a steeper learning curve than adults is particularly intriguing. One possible interpretation of this is that infants who successfully acquire the association may have exhibited a stronger initial response to the air puff, which could have resulted in a steeper initial learning curve. This may be attributed to heightened sensitivity to novelty or a greater focus on establishing new associations during this critical developmental period. The reason for the significant discrepancy in the number of learners and non-learners remains a topic for further investigation.”

4. The authors also did not address my question of whether the variable age range in the adolescent group accounted for their effects. They added a note to the discussion that says someone else could look at this, but this is also an analysis they could run directly.

Response: We thank the reviewer for this important question. Following this suggestion, we analyzed if the distribution in individual blink-per trial ratios could be explained by age differences within the adolescent group. However, age did not significantly predict blinking frequencies on day 1 (beta = -0.02, $t(28)=-0.57$, $p=.572$) nor on day 2 (beta = -0.03, $t(27)=-0.74$, $p=.468$).